# No relationship between fornix and cingulum degradation and within-network decreases in functional connectivity in prodromal Alzheimer's disease

**Therese M. Gilligan**[1,2]*, **Francesca Sibilia**[1,2], **Dervla Farrell**[1,2], **Declan Lyons**[3], **Seán P. Kennelly**[2,4,5], **Arun L. W. Bokde**[1,2]

**1** Discipline of Psychiatry, School of Medicine, Trinity College Dublin, Dublin, Ireland, **2** Trinity College Institute of Neuroscience, Trinity College Dublin, Dublin, Ireland, **3** St Patrick's University Hospital, Dublin, Ireland, **4** Discipline of Medical Gerontology, School of Medicine, Trinity College Dublin, Dublin, Ireland, **5** Memory Assessment and Support Service, Department of Age-related Healthcare, Tallaght University Hospital, Dublin, Ireland

* therese_gilligan@yahoo.co.uk

**Data Availability Statement:** Summary output of processing done via ExploreDTI and Conn, the R code used to analyse said output, and the

## Abstract

### Introduction

The earliest changes in the brain due to Alzheimer's disease are associated with the neural networks related to memory function. We investigated changes in functional and structural connectivity among regions that support memory function in prodromal Alzheimer's disease, i.e., during the mild cognitive impairment (MCI) stage.

### Methods

Twenty-three older healthy controls and 25 adults with MCI underwent multimodal MRI scanning. Limbic white matter tracts–the fornix, parahippocampal cingulum, retrosplenial cingulum, subgenual cingulum and uncinate fasciculus–were reconstructed in ExploreDTI using constrained spherical deconvolution-based tractography. Using a network-of-interest approach, resting-state functional connectivity time-series correlations among sub-parcellations of the default mode and limbic networks, the hippocampus and the thalamus were calculated in Conn.

### Analysis

Controlling for age, education, and gender between group linear regressions of five diffusion-weighted measures and of resting state connectivity measures were performed per hemisphere. FDR-corrections were performed within each class of measures. Correlations of within-network Fisher Z-transformed correlation coefficients and the mean diffusivity per tract were performed. Whole-brain graph theory measures of cluster coefficient and average path length were inspecting using the resting state data.

neuropsychological scores, are available on open science framework: https://osf.io/nvaz7/. This constitutes the minimal underlying data necessary to replicate this study.

**Funding:** This research was supported by a grant from The Meath Foundation that was awarded to A. L.W.B and S.P.K. A.L.W.B. was funded in part by the Science Foundation Ireland (grant number 11/ RFP.1/NES/3194, http://www.sfi.ie/funding/ researcher-database/), and (together with P.G. Mullins and J.P. McNulty) from the European Regional Development Fund via the Interregional 4A Ireland Wales Programme 2007–2013 http:// www.irelandwales.ie//projects/priority_1_theme_2/ neuroskill. The MRI data were accessed from the Lonsdale cluster maintained by the Trinity Centre for High Performance Computing. This cluster was funded through grants from Science Foundation Ireland. The funders had no role in study design, data collection and analysis, decision to publish, or preparation of the manuscript.

**Competing interests:** The authors have declared that no competing interests exist.

## Results & conclusion

MCI-related changes in white matter structure were found in the fornix, left parahippocampal cingulum, left retrosplenial cingulum and left subgenual cingulum. Functional connectivity decreases were observed in the MCI group within the DMN-a sub-network, between the hippocampus and sub-areas -a and -c of the DMN, between DMN-c and DMN-a, and, in the right hemisphere only between DMN-c and both the thalamus and limbic-a. No relationships between white matter tract 'integrity' (mean diffusivity) and within sub-network functional connectivity were found. Graph theory revealed that changes in the MCI group was mostly restricted to diminished between-neighbour connections of the hippocampi and of nodes within DMN-a and DMN-b.

## Introduction

Amnestic mild cognitive impairment (MCI) features a reduced ability to recall episodic events and to form new memories alongside an intact ability to function independently. MCI is a high risk factor for conversion to dementia, in particular Alzheimer's disease (AD), with annual conversion rates of c10-20% [1,2]. AD is thought to evolve slowly and asymptomatically over decades, and current investigations of its earliest stages are mostly focused on MCI. This study aims to probe the relationship between structural and functional brain changes in MCI through use of resting state magnetic resonance imaging (rsMRI) and diffusion weighted imaging (DWI).

The use of in vivo imaging has established that whole-brain cerebral atrophy continues alongside the progression of AD and that these large-scale effects are reflected in the degeneration of a wide-range of behaviours [3]. Initial volume loss is understood to occur in the medial temporal lobe, specifically the hippocampus and entorhinal cortex [4,5]. This profile of cortical atrophy sometimes present in MCI is distinct from normal ageing. However, given the heterogeneity of MCI, it may not be sufficiently specific for a diagnosis of AD [6,7].

AD is widely described as a disconnection syndrome, i.e., it is the disconnection between brain areas that amplifies the cognitive and behavioural decline [8]. For example, it is proposed that early hypometabolism seen in the posterior cingulate cortex reflects *distant* damage in the hippocampal formation more so than *local* neuropathological processes within the posterior cingulate cortex [9,10]. Consistent with the disconnection hypothesis, correlations have been observed in AD between atrophy of white matter tracts related to episodic memory and grey matter atrophy of the hippocampal formation [11].

Synaptic loss and an accumulation of neurofibrillary tangles that disrupt cellular function are possible sources of disconnection [12,13]. Other candidate sources are grey matter atrophy that leads to Wallerian degeneration of white matter, and abnormalities that begin within white matter [14–16]. This last possibility has been investigated used DWI. This technique facilitates the examination of the diffusion of water molecules and can reveal between-group differences in the microstructure of white matter tracts. The causes of such changes can include myelination, axon density, axon diameter, membrane permeability and voxel architecture. It is not, however, possible to specify the exact change and best practice is to provide a range of anisotropy measures [17].

Lancaster et al., [18] found that DWI measures of the hippocampal cingulum and uncinate fasciculus, but not grey matter or white matter of the medial temporal lobe nor DWI measures

of the fornix, predicted a three-year decline in episodic memory in cognitive healthy older people with AD risk factors. These findings add to other reports that white matter damage precedes grey matter atrophy [19–22]. Of note, the white matter tracts most implicated in MCI (the fornix, the cingulum bundle and the uncinate fasciculus) are those that facilitate communication and information transfer to and within medial temporal structures [20,23–28]. That is, white matter damage has been consistently identified in tracts related to regions where initial grey matter volume loss occurs. Nonetheless, it remains an open question as to whether grey or white matter atrophy occurs first or if both degenerate from the outset.

rsMRI assesses the brain's intrinsic functional organisation through measurement of the blood-oxygen-level dependent signal when participants are at rest / not performing any task [29]. Functional connectivity refers to a synchrony in that signal between anatomically distinct regions (measured at rest or during task) that leads to the assumption that those regions are functionally connected [30]. rsMRI has revealed neural networks based on their functional connectivity [31], of these the default mode network (DMN) has been revealed to be widely implicated in MCI (for meta-analysis studies see: [32–34]) and to a lesser extent the limbic network [34]. In MCI functional connectivity between regions within the DMN has been observed as decreased and as enhanced [35,36], while within limbic network has been reported as enhanced [34]. Altered within and between-network functional connectivity has been implicated in other networks (e.g., somatomotor, executive control, dorsal attention) particularly as the disease progresses [37–40]. While functional connectivity enhancements are suggestive of a compensatory mechanism this is not necessarily the case [36]. Increased functional connectivity between the DMN and the frontoparietal network, for example, has been interpreted as a reflection of a difficulty in switching between optimal network behaviours [40].

That finding of increased functional connectivity parallels the lack of segregation between the DMN and frontoparietal networks that have been revealed using a graph theory approach in AD, and to a lesser extent, MCI patients [41]. Graph theory investigations of the brain look at the shape of information transfer at a high level, i.e., the network/connectome level. This focuses on examining how information is segregated within clusters thus facilitating specialisation (functional regions) and how it is integrated across clusters facilitating cross-modal collaboration [42,43]. Published graph theory studies support the idea of AD as a disconnection syndrome given that alterations in both information integration and segregation have been found across different neuroimaging modalities (e.g., EEG: [44,45]; structural: [46–48]; DWI: [49]; rsfMRI: [50,51]).

Previous studies have observed changes in the relationship between structural and functional connectivity in MCI centring on the thalamus [52,53]. This study adds to the literature by investigating constrained spherical deconvolution DWI measures [54] of MCI-targeted white matter tracts (fornix, cingulum bundles, and uncinate fasciculus) and temporal correlation connectivity measures of MCI-targeted functional networks (DMN and limbic), and by examining the relationship between those structural and functional measures. A correlation between these white matter tracts and functional networks would suggest a (non-directional) dependence between their degeneration in MCI. Further, we investigated functional connectivity graph theory measures to provide an alternative and higher level perspective on any MCI related changes.

## Methods

Twenty-eight older adults with amnestic MCI participated in the study. One person was unable to undertake scanning due to undiagnosed claustrophobia, upon data inspection a second person was excluded due to discovery of an undiagnosed historical focal thalamic lesion.

A third person was eliminated due to missing demographic and neuropsychological information. Five MCI people declined to complete the entirety of the neuropsychological testing set (four did not complete the CERAD tests, one did not complete the CERAD or GDS tests)–however, all were successfully scanned. The final sample included 25 MCI participants.

Twenty-three old healthy controls (HC) were recruited from the greater Dublin area via newspaper advertisements. The MCI participants were recruited from Dublin memory clinics in Tallaght University Hospital, St James' Hospital and St Patrick's Hospital. All participants were right-handed and 54–80 years old. Exclusion criteria covered a history of stroke, transient ischaemic attack, heart attack, head injury, neurological illness, psychiatric illness, substance addiction or abuse, abnormal hearing or vision (in presence of necessary correction). The MCI participants were diagnosed by a clinician according to the Petersen criteria [55], i.e., absence of functional decline indicative of dementia but presence of abnormal memory scores relative to age and educational attainment.

The study was conducted in line with the Declaration of Helsinki principles, and it received ethical approval from the St Patrick's University Hospital and Tallaght University Hospital Research Ethics Committees. All participants gave written consent prior to taking part in the study.

## Neuropsychological testing

All participants undertook a health screening questionnaire to assess suitability for scanning. The Consortium to Establish a Registry for Alzheimer's Disease assessment (CERAD, [56]) was used to screen the HCs for undiagnosed age-related cognitive impairment [57]. Further, participants were tested with the mini-mental state examination (MMSE; [58], short-form Geriatric Depression Scale (GDS; [59]), and a Cognitive Reserve Questionnaire (CogR; [60]) before the MRI scan.

Robust independent t-tests were performed on the demographic and neuropsychological tests using the Yuen t-test [61] (bootstrapped and 10% trimmed) from the R package WRS2 [62]. The t-tests revealed that the groups did not differ in age, gender, number of years of education, cognitive reserve or IQ (all $P$s > .05). It was observed that the MCI group performed less well overall on the CERAD battery than the healthy controls. Bonferroni corrected one-sided t-tests revealed statistically worse scores for MMSE, word delay, word recognition (yes), and trail B. For summary details see Tables 1 and 2. Using the CERAD scores *standardised* against age, education and gender norms, a statistically lower performance was observed for MMSE, word delay, trial A and trial B–see Table A in S1 File. The MCI group scored higher on the depression scale ($p$ = .03). Three people in the MCI group and one in the HC group exceed the short-form GDS cut-off score of 5 suggesting depression. Depression is thought to accompany but not precede the development of MCI [63] and to be an additional risk factor in conversion from MCI to AD [64]. MCI depression score did not correlate with those cognitive measures that survived correction for multiple comparisons. There were, however, correlations with the *standardised* measures of fluency ($p$ = .002) and naming ($p$ = .032) that unexpectedly indicated increasing cognitive scores with increasing GDS score–see Figs A and B in S1 File for further details.

## MRI Data acquisition

All data was acquired on a 3.0 Tesla Philips Achieva MR system (Best, The Netherlands) with an eight channel head coil. A high-resolution 3D T1-weighted anatomical image was acquired: Echo time (TE) = 3.9 ms, repetition time (TR) = 8.4 mm, field of view (FOV) = 230 mm, slice thickness = 0.9 mm, voxel size = 0.9 m x 0.9 mm x 0.9 mm, total scan time was 5 min 46 s. For

**Table 1. Demographic details.**

| Profile | HC | MCI | Statistic | P-value* |
|---|---|---|---|---|
| Gender (M/F %) | 65/35 | 44/56 | $X^2 = 2.17$ | 0.141 |
| Age | 69 ± 2.66 | 68 ± 6.28 | $Y_t = -0.7369$ | 0.441 |
| Education | 14 ± 3.85 | 12.6 ± 2.75 | $Y_t = -1.2592$ | 0.1995 |
| IQ | 118 ± 7.87 | 114 ± 6.72 | $Y_t = -1.6943$ | 0.107 |

Raw mean + SD

* *p*-value: 2-sided, uncorrected

the DWI acquisition whole-brain high angular resolution diffusion imaging (HARDI) data was acquired using a parallel sensitivity encoding (SENSE) approach [65] with a reduction factor of two. It was acquired using single-shot spin echo-planar imaging (EPI): TE = 81 ms, TR = 14,556 ms, FOV = 224 mm, matrix 112 x 112, voxel size = 2 mm x 2 mm x 2 mm, and 65 slices with 2 mm thickness and no gaps, total scan duration was 18 min 6 s. Diffusion gradients were applied in 61 isotropically distributed orientations with $b$ = 2000 s/mm$^2$, four images with $b$ = 0 s/mm$^2$ were also acquired.

Data acquisition for the rsMRI lasted 7 min. A T2*-weighted echo-planar imaging sequence with TE = 27 ms and TR = 2000 ms was used to acquire the blood oxygenation dependent (BOLD) signal. Two hundred and ten volumes of data were acquired, voxel size = 3.5 mm x 3.5 mm x 3.85 mm with a 0.35 mm gap between slices. Thirty-nine slices, covering the entire brain, were imaged per volume. Slices were acquired in an interleaved sequence from an inferior to superior direction. During this scan, participants were instructed to fixate on a red cross-hair in the centre of a screen behind the MRI scanner, it was visible via a mirror.

**Pre-processing and data extraction.** T1-w: T1-w images were oriented to standard position (FSL; [66]), labelling was verified and images were visually assessed for quality and incidental findings. FSL-ANAT segmentation method was used to extract tissue volumes (grey matter, white matter and cerebrospinal fluid) in order to estimate intra-cranial volume.

DWI: The Philips diffusion-weighted images were converted to Nifti files using dcm2niix [67]. They were then pre-processed using ExploreDWI version 4.8.4 [68]. This included

**Table 2. Neuropsychological tests.**

| Measure | HC | MCI | Statistic ($Y_t$) | P-value* |
|---|---|---|---|---|
| GDS | 1.04 ± 1.72 | 2.25 ± 2.07 | 2.53 | 0.030 |
| Cognitive Reserve | 17.2 ± 4.99 | 15.68 ± 4.86 | -0.9464 | 0.347 |
| **CERAD:** | | | | |
| MMSE | 28.9 ± 0.949 | 26.7 ± 2.34 | -3.5309 | 0.001 |
| Fluency | 17.7 ± 3.55 | 15.3 ± 4.87 | -1.9598 | 0.056 |
| Naming | 14.7 ± 0.559 | 12.9 ± 2.25 | -2.9748 | 0.016 |
| Word Delay | 8.6 ± 1.27 | 4.21 ± 2.39 | -6.5326 | 0.000 |
| Word Recognition Y | 9.9 ± 0.209 | 8.65 ± 1.53 | -3.6551 | 0.0095 |
| Word Recognition N | 10 ± 0 | 9.60 ± 0.821 | -1.51 | 0.134 |
| Praxis | 10.5 ± 0.846 | 10.1 ± 0.898 | -1.5223 | 0.100 |
| Trail A | 35.9 ± 7.68 | 48.9 ± 17.7 | 2.4595 | 0.025 |
| Trail B | 74.5 ± 21.0 | 139 ± 58.4 | 4.9546 | 0.0005 |

Raw score mean + SD

* *p*-value: 2-sided, uncorrected.

converting the Philips bval and bvec files to a bmatrix (txt) file. B0 fieldmaps of the bmatrix and Nifti files were brought to the beginning of the images as appropriate. The Nifti files were made ExploreDWI compatible, gradients were permuted and flipped as required and the files converted to matlab image files. Corrections for subject motion, eddy current and EPI were made in one step using the Robust approach (Rekindle linear), during which the images were registered to their respective ExploreDWI-compatible T1-w files, using the methods described elsewhere [69,70].

Whole-brain tractography was run on the corrected files using a constrained spherical deconvolution method [71], a deterministic approach. This method can account for complex white matter orientation such as crossing fibres [72] and has previously detected changes in MCI and AD [54]. Using every voxel as a seed point, and in increments of 1 mm, the principal diffusion orientation at each point was estimated. Tracking moved along the direction that subtended the smallest angle to the current trajectory. A trajectory was followed until the scaled height of the fibre orientation density function peak dropped below 0.1, or the direction of the pathway changed through an angle of no more than 30˚.

Following whole-brain tractography, the different tracts were extracted by manually drawing several regions of interest (ROIs) defined according to published methods for the fornix [73,74], the subgenual and retrosplenial branches of the cingulum [75], the parahippocampal branch of the cingulum [76], and the uncinate fasciculus [73]. See Fig 1 for further details on the placement of the ROIs.

Using an atlas-based tractography approach the ROIs were located on three template individuals and then applied to participants pre-matched to each template-individual. Each template individual was chosen to represent those with small, medium or large ventricles, identified as such based on visual inspection, and classification, of the entire group of participants. The atlas-based tractography approach spatially transforms the ROIs, manually drawn on the templates, to the other subjects' native space. This ensures consistency in the identification of the tracts. The use of three templates did not always overcome inter-subject variability issues, as was evident in missing or slight tracts or tracts with excessive spurious streamlines. In such cases (17% for fornix, 8% for cingulum bundles, 44% for uncinate fasciculus) the ROIs were manually drawn and adjustments were made as necessary. In all cases tidying of the tracts was achieved by the application of one or several NOT gates.

Resting State fMRI: The rsMRI data were processed using the Conn v18a toolbox [77] run in SPM v8 [78]. A default MNI152-space data pre-processing template was applied consisting of: functional realignment and unwarping, slice-timing correction, structural segmentation and normalisation, functional normalisation, outlier detection, and smoothing. Segmentation and normalisation steps were supported by the acquired structural T1-w images. Structural target resolution was set at 1 mm isotropic, functional target resolution was set at 2 mm isotropic. Smoothing was done using a 4 mm full-width-at-half-maximum Gaussian kernel.

Using Conn default settings, potential confounding effects removed from the BOLD signal using linear regression were: white matter and cerebrospinal fluid time series (5 regressors each, CompCor approach, [77], scrubbing (invalid scans: M = 6.71, SD = 12.3, range 0–54, no difference between groups, $Y_t = 1.09$, $p = .309$ ), realignment (6 motion parameters and 6 first-order temporal derivatives) and the effect of rest. Band-pass filtering (0.01–0.08 Hz) and linear detrending were included in this denoising step.

The BOLD signal time series was extracted from sub-cortical regions and cortical networks known to be implicated in early Alzheimer's Disease [79]. The hippocampus and thalamus were identified using the FSL Harvard-Oxford Atlas (http://fsl.fmrib.ox.ac.uk/fsl/fslwiki/Atlases). Using the 2mm 400 region cortical atlas [80] two limbic (a and b) and three default mode sub-networks (a, b, and c) per hemisphere were identified. These atlas parcellations

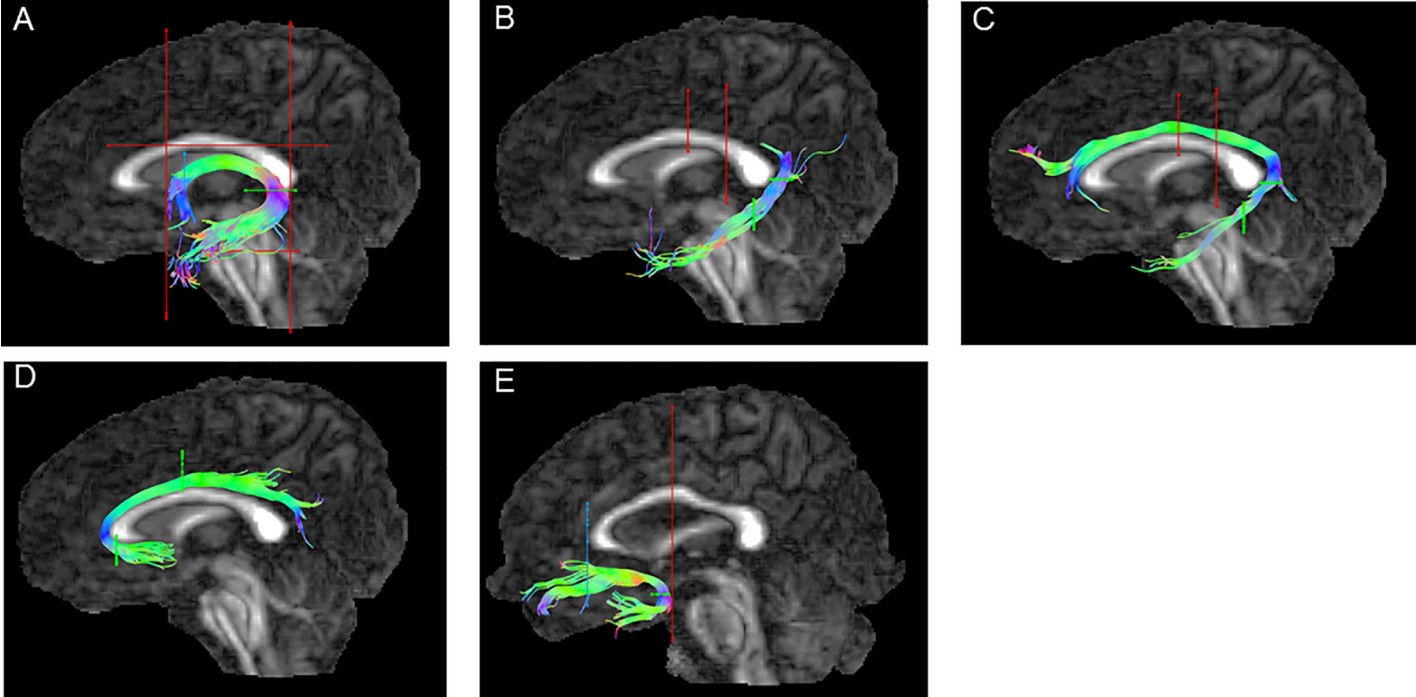

**Fig 1. Region of interest placement for each white matter tract.** Within ExploreDTI the blue lines indicate a 'seed/or' gate, the green line an 'and' gate, red lines indicate 'not' gates. Tracts are shown for the left hemisphere in a template subject. A) fornix, B) parahippocampal cingulum, C) retrosplenial cingulum, D) subgenual cingulum, and E) uncinate fasciculus.

were computed from functional connectivity patterns. The sources of the functional connectivity signals within each sub-network are detailed in Fig 2. Centroid co-ordinates for the parcels are presented in S1 File Table L. All atlases and the pre-processed resting state images were in MNI152 space. A weighted sum time series method was used to extract the BOLD time series signals for each ROI. Connectivity measures were calculated using a haemodynamic response function weighted general linear model for bivariate correlations, set at a default 0.25 threshold. Using the Conn toolbox this step outputs Fisher Z-transformed correlation coefficients per ROI-to-ROI pairing for each participant.

To estimate global measures of graph theory the above steps for functional connectivity correlations were also followed, but this time the signal was extracted from the whole brain. All sub-cortical and cerebellar segmentations were defined using the FSL Harvard-Oxford Atlas (http://fsl.fmrib.ox.ac.uk/fsl/fslwiki/Atlases), and all cortical ROIs were extracted from the Schaefer et al., [80] 400 ROI atlas. Measures of cluster coefficient and average path length were inspected at a cost (sparsity) level of 0.15.

## Analyses

Due to the general concern regarding small sample sizes meeting the assumptions of the general linear model, a robust approach was undertaken [81]. The R package robustbase [82] was used to perform robust multiple linear regressions. Its lmrob function fits a model based on an M-estimator using iteratively reweighted least squares estimation [83]. Linear regressions of diffusion-weighted measures and resting-state connectivity measures conditional upon group were performed per hemisphere. All tests controlled for age, education and gender. The relationship between depression and cognitive function is complicated and some depressive

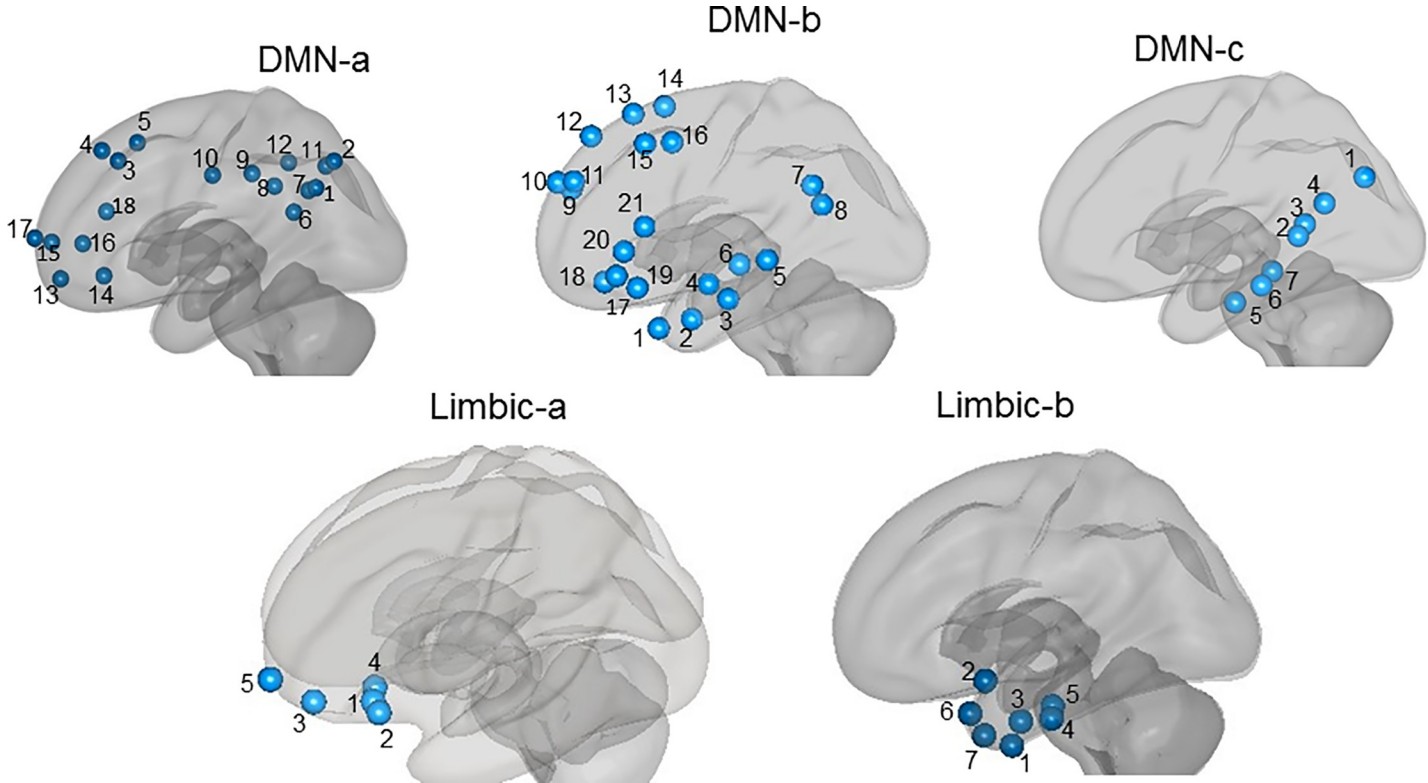

**Fig 2. The three parcellations of the DMN and two parcellations of the limbic networks as delineated by the Schaefer et al., 2018 atlas [80].** *DMN-a*: 1–2) inferior parietal lobule 1&2, 3–5) dorsal prefrontal cortex 1–3, 6–12) posterior cingulate cortex 1–7, 13–18) medial prefrontal cortex 1–6. *DMN-b*: 1–6) temporal cortex 1–6, 7–8) inferior parietal lobule 1&2, 9–14) dorsal prefrontal cortex 1–6, 15–16) left prefrontal cortex 1&2, 17–21) ventral prefrontal cortex 1–5. *DMN-C*: 1) inferior parietal lobule 1, 2–4) retrosplenial cortex 1–3, 5–7) parahippocampal cortex 1–3. *Limbic a*: 1–5) orbitofrontal cortex 1–5. *Limbic-b*: 1–7) temporal pole 1–7.

elderly will not convert to dementia [84]. However, we chose not to include depression as a control covariate given that its presence may reflect dementia pathology [85]. That is, controlling for depression would run the risk of removing relevant explanatory variance. Further, in this group it would both reduce sample size as one participant declined to complete the GDS and risk over-fitting the model. In any case, in this cohort depression did not correlate with worsening measures of cognition (see Fig B in S1 File). FDR-corrections were performed within each class of measures. With the exception of the graph theory analysis, the relevant measures were extracted from their respective processing package and analysed within R version 3.5.0 [86]. Additional R packages used were dpylr [87], ggplot2 [88], stringr [89].

The diffusion metrics extracted for analysis were fractional anisotropy (FA), mean diffusivity (MD), axial diffusivity (Da) and radial diffusivity (Dr) and tract volume. Lower values of FA, and higher values of MD, Da and Dr, were predicted in the MCI compared to the HC group [90]. Tract volume was divided by total intracranial volume in native space prior to statistical testing (metric = $mm^3$). Tract volume was predicted to be lower in the MCI group. P-values were adjusted to take into account the directional hypotheses.

Fisher Z-transformed bivariate correlation coefficients of BOLD signal time series were averaged/calculated within the chosen networks, between the networks, and between the networks and the subcortical ROIs, per hemisphere. No directional predictions were made given that both decreased and increased connectivity have previously been observed within AD samples [46]. (Within hemisphere only analyses were conducted, across hemisphere connectivity was not inspected due to sample size).

Within group and per hemisphere, robust percentage bend correlations of the within-network Fisher Z-transformed bivariate correlation coefficients and the normalised MD values of each tract were performed using the using the WRS2 package [62]. These inspected associations were not limited to known structural connectivity (e.g. fornix and DMN-c) based on two assumptions. First, human structural connectivity is not yet so precisely delineated that exclusive connections are assured (e.g., the uncinate fasciculus likely facilitates connections with various networks such as limbic-a, limbic-b networks, DMN-a, and DMN-c). Second, taking an agnostic approach allowed possible secondary, or downstream, effects of tract degeneration to be considered [91,92]. However, in order to constrain the number of analyses, these correlations were focused on the MD metric, as it is understood to be the diffusion measure most sensitive to AD changes [24,93]. Associations of tract MD with between-network functional connectivity (over 200 possible correlations) were not inspected in order to constrain the analysis.

Between-group rsMRI graph theory measures of cluster coefficient and average path length were inspected in order to provide a global overview of functional connectivity brain changes. Cluster coefficient is a local measure that examines the number of nearest neighbours of a node as a proportion of the maximum possible number of connections. From the connectome perspective it measures segregation—the efficiency of information transfer at a local scale. Path length is a global measure of integration; it quantifies the overall routing efficiency of a network by examining the average minimum number of connections that link any two nodes of a network [42]. These analyses were conducted within the Conn toolbox. Cost (sparsity) was set at 0.15, no directional prediction was made. The tests controlled for age, education and gender and FDR-corrections were applied to follow-up tests.

## Results

### DWI

All diffusion measures of the fornix showed evidence of degeneration in the MCI group: MD, Da and Dr were comparatively increased in the left (MD: $t$ = 4.21, $p_{cor}$ = .0006, Da: $t$ = 5.32, $p_{cor}$ = .00005, Dr: $t$ = 3.74, $p_{cor}$ = .002) and the right fornix (MD: $t$ = 5.06, $p_{cor}$ < .0001, Da: $t$ = 5.59, $p_{cor}$ = .00005, Dr: $t$ = 4.65, $p_{cor}$ = .0002), and FA was comparatively decreased in the left ($t$ = -2.78, $p_{cor}$ = .019) and right ($t$ = -2.42, $p_{cor}$ = .035) fornices. In the MCI compared to the HC group, MD and Da were significantly increased in the left parahippocampal cingulum (MD: $t$ = 2.72, $p_{cor}$ = .019, Da: $t$ = 3.02, $p_{cor}$ = .012), and MD and Dr were significantly increased in the left retrosplenial cingulum (MD: t = 3.08, $p_{cor}$ = .011, Dr: $t$ = 2.43, $p_{cor}$ = .035) and in the left subgenual cingulum (MD: $t$ = 3.08, $p_{cor}$ = .011, Dr: $t$ = 2.77, $p_{cor}$ = .019).

All other measures were in the expected direction (i.e., comparatively decreased FA and tract volume and increased MD, RD and AD in the MCI group) but either did not survive corrections for multiple comparisons or were not significant at the uncorrected level. There were two exceptions to this, Da values were the same for each group in the left uncinate fasciculus and were comparatively decreased in the right uncinate fasciculus in the MCI group. Details of the MD values are presented in Table 3, Figs 3 and 4 display the DWI metrics for the fornix and parahippocamapl cingulum, see S1 File for other diffusion measure descriptives (Tables B-F) and graphs (Figs C-E).

### rsMRI

In MCI compared to HC, statistically smaller within-network connectivity was found in the DMN-a in the left ($t$ = -3.38, $p_{cor}$ = .010) and right ($t$ = -3.75, $p_{cor}$ = .005) hemispheres. See Table 4.

**Table 3. MD values per structure per hemisphere.**

| Structure | Hemi | HC | MCI | Estimate | t-statistic | p-value * |
|---|---|---|---|---|---|---|
| Fornix | LH | 0.00120 ± 0.00008 | 0.00132 ± 0.00014 | 0.000139 | 4.210 | 0.000064 |
| | RH | 0.00118 ± 0.00008 | 0.00127 ± 0.00008 | 0.000100 | 5.056 | .0000042 |
| Para-hippocampal cingulum | LH | 0.00072 ± 0.00003 | 0.00075 ± 0.00005 | 0.0000187 | 2.721 | 0.00467 |
| | RH | 0.00072 ± 0.00003 | 0.00074 ± 0.00005 | 0.0000104 | 1.007 | 0.15950 |
| Retrosplenial cingulum | LH | 0.00067 ± 0.00002 | 0.00069 ± 0.00002 | 0.0000213 | 3.084 | 0.00178 |
| | RH | 0.00067 ± 0.00002 | 0.00068 ± 0.00003 | 0.0000148 | 2.052 | 0.02315 |
| Subgenual cingulum | LH | 0.00069 ± 0.00002 | 0.00070 ± 0.00002 | 0.00001516 | 3.078 | 0.00181 |
| | RH | 0.00069 ± 0.00002 | 0.00069 ± 0.00002 | 0.00000946 | 1.706 | 0.04765 |
| Uncinate fasciculus | LH | 0.00070 ± 0.00002 | 0.00071 ± 0.00004 | 0.00000444 | 0.806 | 0.21225 |
| | RH | 0.00072 ± 0.00002 | 0.00072 ± 0.00003 | 0.00000254 | 0.336 | 0.36900 |

Hemi = hemisphere, LH = left hemisphere, RH = right hemisphere

*p-values are uncorrected, 1-sided

In MCI compared to HC, statistically smaller between-network connectivity was found between DMN-a and DMN-c, between DMN-a and the hippocampus, and between DMN-c and the hippocampus in both the left hemisphere (respectively: $t$ = -4.63, $p_{cor}$ = .002; $t$ = -4.21, $p_{cor}$ = .003; $t$ = -3.99, $p_{cor}$ = .003) and the right hemisphere (respectively: $t$ = -3.81, $p_{cor}$ = .005; $t$ = -3.31, $p_{cor}$ = .011; $t$ = -3.52, $p_{cor}$ = .008). In the right hemisphere only in MCI compared to HC, statistically smaller between-network connectivity was found between DMN-c and the thalamus ($t$ = -3.02, $p_{cor}$ = .021) and between DMN-c and the limbic-a network ($t$ = -4.01, $p_{cor}$ = .003). See Table 5, and Figs 5 and 6, for results mentioned here and Table G in S1 File for the complete set.

## DWI and rsMRI

Robust correlations between the normalised MD of each of the five tracts and each of the five within-network Fisher Z-transformed correlations were conducted per hemisphere and group (total of 5x5x2 correlations). At the uncorrected level two associations were significant–between the right parahippocampal cingulum and Limbic-a in the MCI group ($r$ = -0.525, $p$ = .007), and between the left retrosplenial cingulum and Limbic-a in the healthy controls ($r$ = -0.524, $p$ = .010)–but neither survived corrections for multiple comparisons. Given these results, no between-group comparisons were carried out. See Figs F-J in S1 File for full details.

## Graph theory

Graph theory analyses, at sparsity level of 0.15, revealed a significant between-group difference in the cluster coefficient measure. This was higher in the HC (M = 0.492, SD = 0.027) compared to the MCI (M = 0.474, SD = 0.033) group (**b** = .02, $t$ = -2.53, $p_{uncor}$ = .015). This difference was driven by 18 ROIs—8 in DMN-a (5 RH, 3 LH), 5 in DMN-b (2 RH, 3LH), both hippocampi, two ROIs in the LH somatomotor network and one in the RH salient ventral attention network–all of which survived FDR-correction. See Fig 7 and Table 6 for details. The difference in average path length between the two groups did not reach statistical significance (b = .03, $t$ = -1.94, $p_{uncor}$ = .059; HC: M = 2.08, SD = 0.049; MCI: M = 2.06, SD = 0.06).

Given the challenges of defining a-priori an appropriate network cost level [94], and following feedback, we additionally inspected the graph theory measures at sparsity levels of .10, .20 and .25. There were significant differences between the two groups in the cluster coefficient measure for sparsity levels of 0.10, 0.20 and 0.25 (uncorrected for multiple comparisons across

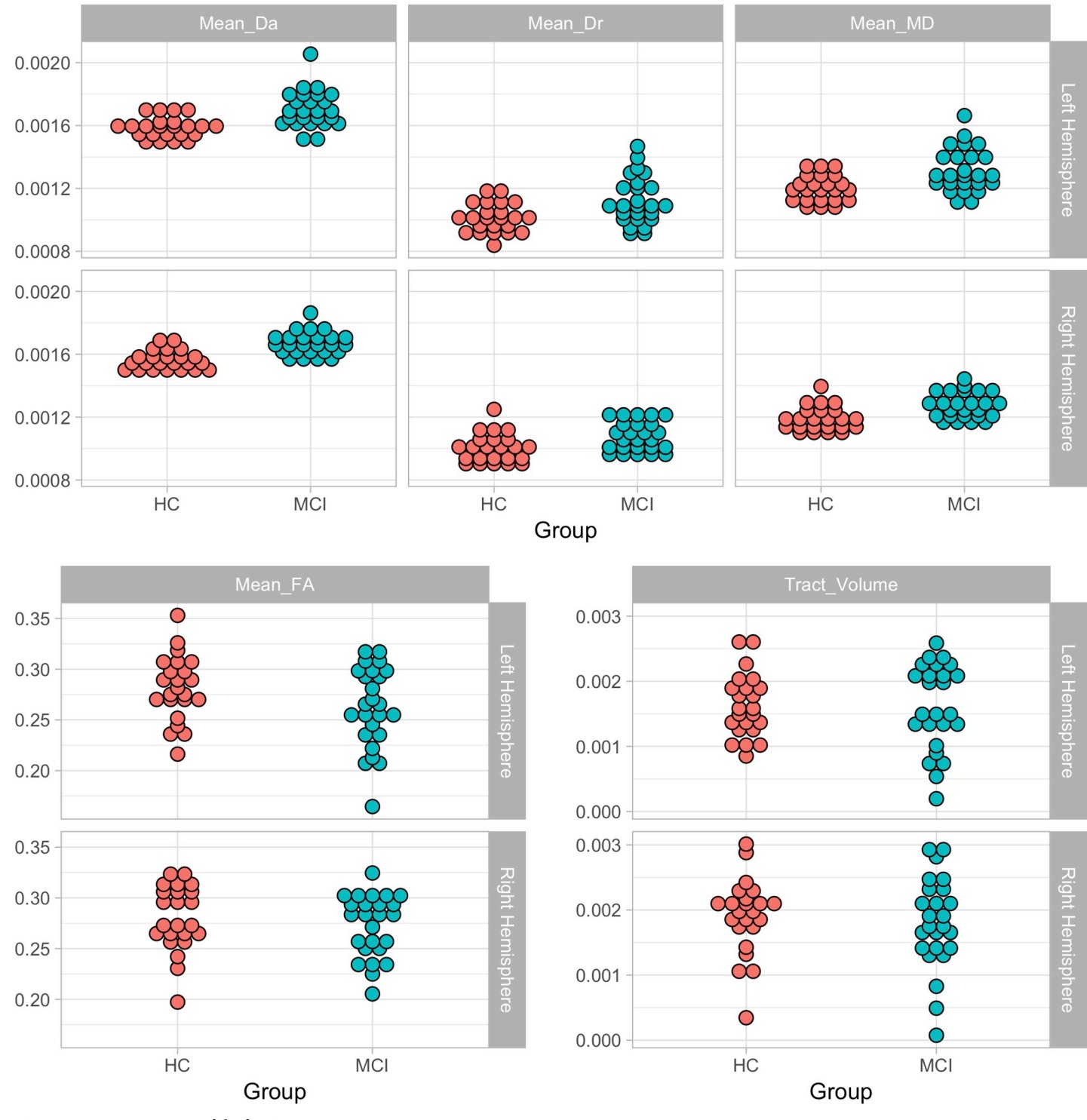

**Fig 3. Mean DWI measures of the fornix.**

levels). As per cost level 0.15, the cluster coefficient measures were higher in the HC compared to the MCI group. At level 0.10 the difference was driven by 16 ROIs (12 from the DMN and hippocampus– 10 of which drive the group differences at sparsity level 0.15). At level 0.20 no

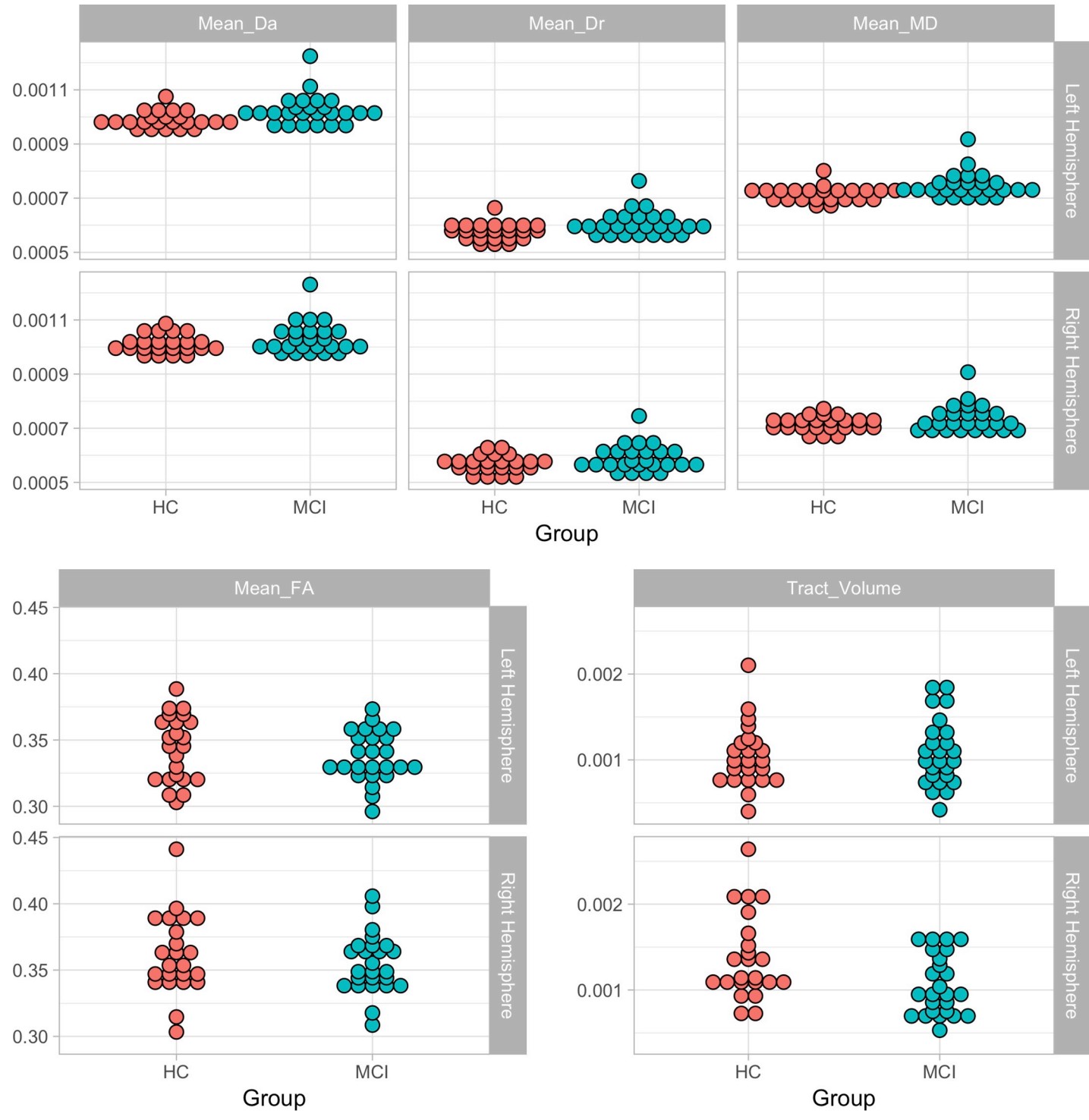

**Fig 4. Mean DWI measures of the parahippocampal cingulum.**

individual ROI survived correction for multiple comparisons. At level 0.25 the difference was driven exclusively by LH temporal area 3 of the DMN_b (it was also a driver of differences at levels 0.10 and 0.15). Full details are provided in S1 File. There was a significant difference in

**Table 4. Within network functional connectivity.**

| Measure | HC | MCI | Estimate | t-statistic | P-value* |
|---|---|---|---|---|---|
| *Within LH:* | | | | | |
| DMN-a | 0.506 ± 0.078 | 0.416 ± 0.133 | -0.100 | -3.379 | 0.00156 |
| DMN-b | 0.352 ± 0.105 | 0.291 ± 0.062 | -0.058 | -2.552 | 0.01435 |
| DMN-c | 0.511 ± 0.129 | 0.421 ± 0.118 | -0.085 | -2.535 | 0.0150 |
| Limbic-a | 0.413 ± 0.126 | 0.404 ± 0.178 | -0.002 | -0.044 | 0.965 |
| Limbic-b | 0.251 ± 0.098 | 0.265 ± 0.079 | -0.009 | -0.315 | 0.754 |
| *Within RH:* | | | | | |
| DMN-a | 0.485 ± 0.109 | 0.381 ± 0.122 | -0.119 | -3.751 | 0.000522 |
| DMN-b | 0.302 ± 0.105 | 0.275 ± 0.096 | -0.033 | -1.194 | 0.239 |
| DMN-c** | 0.462 ± 0.132 | 0.368 ± 0.108 | -0.093 | -2.568 | 0.0135 |
| Limbic-a | 0.349 ± 0.142 | 0.353 ± 0.122 | 0.006 | 0.147 | 0.884 |
| Limbic-b | 0.225 ± 0.074 | 0.257 ± 0.088 | 0.018 | 0.679 | 0.501 |

* *p*-values are uncorrected 2-sided

** No covariates were included due to non-convergence of model

the average path length between the groups at sparsity level of 0.10. As per sparsity level 0.15, this average path length difference was not present at levels 0.20 and 0.25. Further details are provided in S1 File.

## Discussion

This study employed multimodal imaging to investigate relationships between brain regions known to be impaired early in AD. The microstructures of five relevant white matter tracts were analysed using DWI measures. rsMRI was used to investigate functional connectivity across implicated parcellated networks and sub-cortical regions. Relationships between these respective measures of tract 'health' and connectivity 'health' were assessed. Finally, to provide a high-level overview, global measures of graph theory were extracted from rsMRI correlations of the entire brain.

MCI-related disturbances in white matter structure were found in the fornix, in the left parahippocampal cingulum, the left retrosplenial cingulum and the left subgenual cingulum.

**Table 5. Highlights of between network functional connectivity.**

| Measure | HC | MCI | Estimate | t-statistic | P-value* |
|---|---|---|---|---|---|
| **LH** | | | | | |
| DMN -a & -c | 0.352 ± 0.075 | 0.251 ± 0.111 | -0.108 | -4.63 | 0.0000336 |
| DMN-a & Hippocampus | 0.238 ± 0.117 | 0.114 ± 0.125 | -0.149 | -4.21 | 0.000127 |
| DMN-c & Hippocampus | 0.409 ± 0.096 | 0.262 ± 0.138 | -0.157 | -3.99 | 0.000254 |
| **RH** | | | | | |
| DMN -a & -c | 0.328 ± 0.089 | 0.217 ± 0.092 | -0.116 | -3.81 | 0.000438 |
| DMN-a & Hippocampus | 0.208 ± 0.123 | 0.104 ± 0.156 | -0.128 | -3.31 | 0.00192 |
| DMN-c & Hippocampus | 0.326 ± 0.124 | 0.186 ± 0.137 | -0.157 | -3.52 | 0.00102 |
| DMN-c & Thalamus** | 0.147 ± 0.091 | 0.039 ± 0.144 | -0.109 | -3.02 | 0.00408 |
| DMN-c & Limbic-a | 0.133 ± 0.085 | 0.080 ± 0.069 | -0.082 | -4.01 | 0.00024 |

*p*-values are uncorrected 2-sided

** No covariates were included due to non-convergence of model

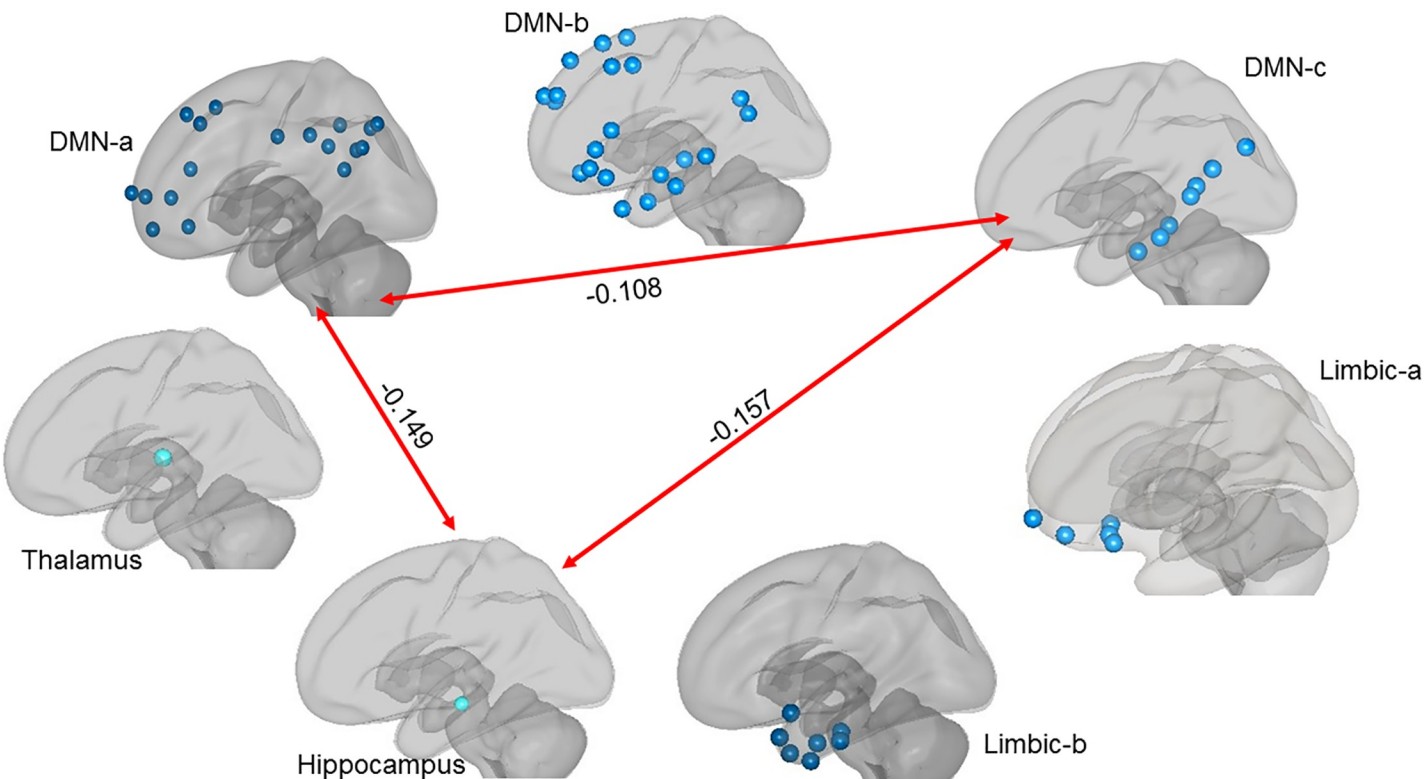

**Fig 5. The red lines with beta values indicate where between region functional connectivity was higher in HC compared to MCI in the left hemisphere.**

No such changes were found in the uncinate fasciculus. Functional connectivity decreases were observed in the MCI group within the DMN, but not the limbic, sub-networks. Functional connectivity was decreased in the MCI group, between the hippocampus and sub-areas a and c of the DMN, between DMN-c and DMN-a, and, in the right hemisphere only, between DMN-c and both the thalamus and limbic-a. No relationships between white matter tract 'health' (MD metric) and within sub-network functional connectivity were detected. The observed region-of-interest functional connectivity disturbances were broadly reflected in the whole-brain cluster coefficient measure of graph theory. It revealed that impact of the putative AD-related pathology in the MCI group was observed in, and mostly restricted to, between-neighbour connections of the hippocampi and of nodes within DMN-a and DMN-b.

White matter tractography studies of MCI and early stage AD have found that absolute measures of diffusivity (MD, Dr, Da) are more sensitive detectors of pathology compared to ratio measures such as FA, which reflect changes in the shape of the diffusion ellipsoid [24,25,93,95,96]. This pattern of diffusion metrics is reflected in the present results, with only absolute diffusivity measures reaching (uncorrected and corrected) statistical significance in the cingulum branches.

Damage in the left hippocampal cingulum is the most consistent finding across different types of DWI analysis and stage of MCI [24]. In the present study white matter changes in the cingulum reached corrected statistical significance in the left hemisphere only. Lateralised tract damage has been previously reported, e.g., increased MD in the right posterior cingulate fasciculus in MCI [97]; increased MD in the left cingulum bundle in MCI [98]; decreased FA in left parahippocampal cingulum in MCI [99], increased FA in the left anterior temporal lobe in AD [100], decreased Dr in left uncinate fasciculus in AD [101]. Nonetheless, bilaterally the

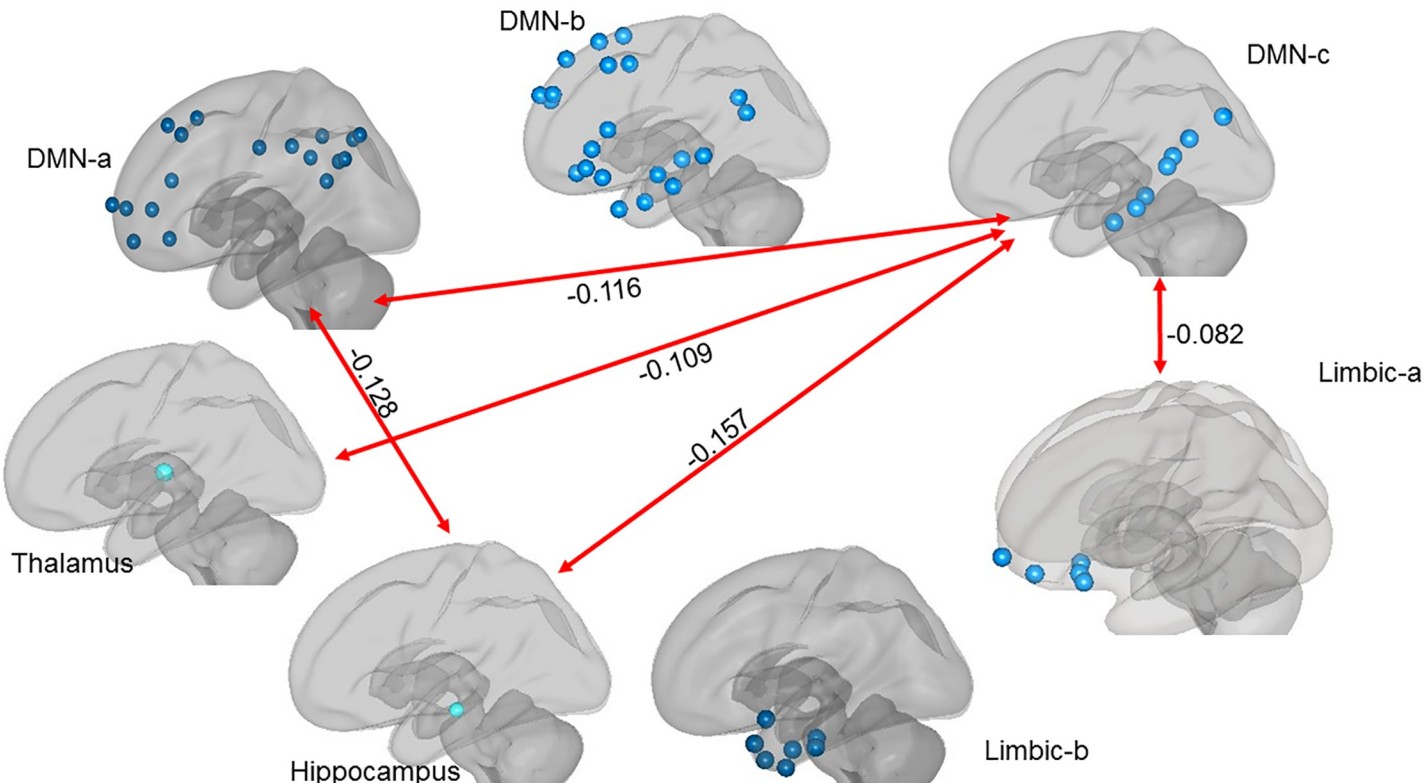

**Fig 6. The red lines with beta values indicate where between-region functional connectivity was higher in HC compared to MCI in the right hemisphere.**

overall white matter changes found in the current paper were in the expected direction [102,103]. This directional effect is true also of the uncinate fasciculus, although, unlike published studies [97,104] we did not find a statistically significant MCI-related change.

The rsMRI results reveal that the strength of *within*-sub-network functional connectivity is reduced in the DMN (in DMN-a only at corrected p-value) but not in the limbic networks in the MCI group. DMN-c (retrosplenial cortex, parahippocampal cortex and inferior parietal nodes) and the hippocampus were implicated in the observed decreases in *between*-network connection strength in the MCI group. This reflects the findings that the medial temporal lobe is the originating grey matter site of damage in AD [4,5]. No evidence of increased connectivity (putatively compensatory or reflective of switching difficulty) was found as has been reported elsewhere [40].

In the current sample global white and grey matter atrophy are present in the MCI group– (see Table K and Fig K in S1 File). However, while both the white matter and the connectivity strength analyses reveal insults to the system, no relationship between the different types of damage was apparent. White matter and grey matter damage in AD may or may not occur independently [19,105]. [106] observed white matter network alterations in preclinical AD that preceded cortical atrophy and hypoglucose metabolism. The retrogenesis hypothesis has been suggested as a putative mechanism for that order of attack [107,108]. However, white matter damage may also be secondary to grey matter damage via Wallerian degeneration [14– 16]. The lack of relationship between structural insult and functional dysconnectivity seen here may be indicative of independent and non-interacting degenerative processes during MCI (in this sample) or it may be related to limitations of our chosen analysis.

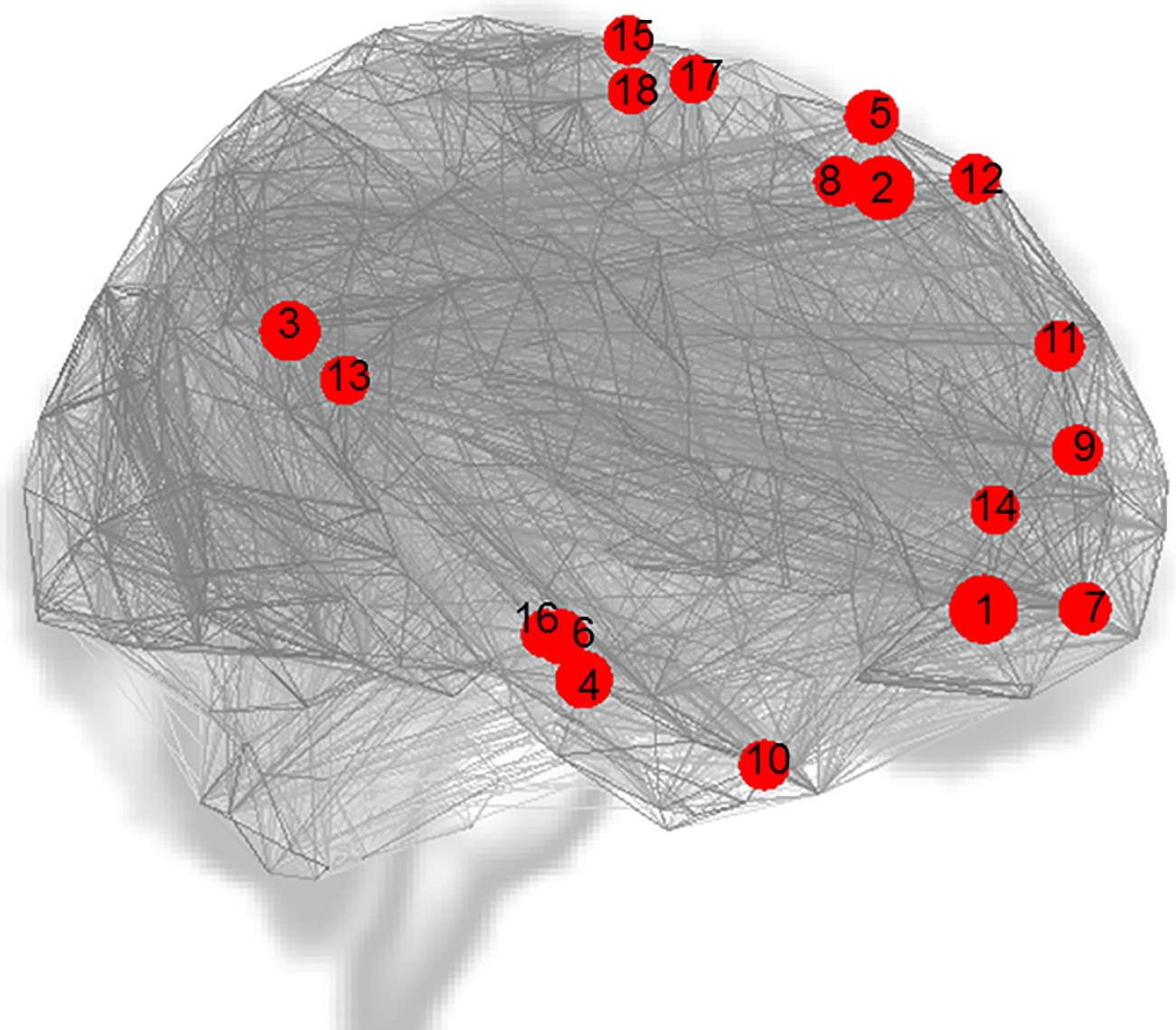

**Fig 7. Points represent regions where the cluster coefficient was significantly larger in HC compared to MCI, at sparsity level of 0.15.** Both hemispheres are presented on the same view. 1) RH DMN-a PFCm_1; 2) RH DMN-a PFCd_2; 3) LH DMN-a PCC_2; 4) LH DMN-b Temp 3; 5) LH DMN-b PFCd_5; 6) RH hippocampus; 7) LH DMN-a PFCm_1; 8) LH DMN-a PFCd_3; 9) RH DMN-a PFCm_4; 10) RH DMN-b AntTemp_1; 11) LH DMN-b PFCd_1; 12) RH DMN-b PFCd_4; 13) RH DMN-a PCC_1; 14) RH DMN-a PFCm_3; 15) LH SomMot_a 16; 16) LH hippocampus; 17) RH SalVentAttn_a ParMed_7; 18) LH SomMot_a 12. PFC = prefrontal cortex; d = dorsal; m = medial; Temp = temporal; Ant = anterior; SomMot = somatomotor; SalVentAttn = salient ventral attentional.

Specifically, it may be that the functional parcellations used are too coarse (e.g., the DMN-a is composed of 18 functionally connected regions) to probe structural and functional relationships. It is also possible that combinations of damage to different tracts, rather than individual tract damage as examined here, are related to within-network dysconnectivity, especially at the early stages of degeneration [109–111]. We did not examine such multivariate relationships due to sample sizes constraints. In addition, it is possible that damage in non-examined white matter tracts, including superficial and short range association fibres, may influence functional connectivity within the DMN in MCI [112,113]. It is also possible that, notwithstanding the

**Table 6. Graph theory cluster coefficient, sparsity level of 0.15.**

| Measure | HC | MCI | Estimate | t-statistic | P-value* |
|---|---|---|---|---|---|
| **LH** | | | | | |
| DMN-a PCC 2 | 0.569 ± 0.045 | 0.510 ± 0.067 | 0.07 | -4.17 | 0.000146 |
| DMN-b Temp 3 | 0.564 ± 0.054 | 0.481 ± 0.096 | 0.09 | -3.98 | 0.000262 |
| DMN-b PFCd 5 | 0.540 ± 0.087 | 0.454 ± 0.089 | 0.10 | -3.79 | 0.000458 |
| DMN-a PFCm 1 | 0.536 ± 0.065 | 0.471 ± 0.066 | 0.07 | -3.65 | 0.000713 |
| DMN-a PFCd 3 | 0.525 ± 0.073 | 0.469 ± 0.071 | 0.07 | -3.53 | 0.001002 |
| DMN-b PFCd 1 | 0.543 ± 0.065 | 0.471 ± 0.075 | 0.08 | -3.49 | 0.001117 |
| SomMoTA 16 | 0.546 ± 0.089 | 0.458 ± 0.104 | 0.10 | -3.42 | 0.001383 |
| Hippocampus | 0.501 ± 0.094 | 0.429 ± 0.081 | 0.09 | -3.35 | 0.001687 |
| SomMoTA 12 | 0.544 ± 0.089 | 0.466 ± 0.089 | 0.09 | -3.29 | 0.002004 |
| **RH** | | | | | |
| DMN-a PFCm 1 | 0.532 ± 0.062 | 0.451 ± 0.071 | 0.09 | -4.77 | 0.000022 |
| DMN-a PFCd 2 | 0.559 ± 0.061 | 0.466 ± 0.080 | 0.10 | -4.46 | 0.000058 |
| Hippocampus | 0.514 ± 0.123 | 0.421 ± 0.093 | 0.11 | -3.78 | 0.000472 |
| DMN-a PFCm 4 | 0.542 ± 0.062 | 0.477 ± 0.071 | 0.07 | -3.52 | 0.001023 |
| DMN-b Ant Temp 1 | 0.541 ± 0.063 | 0.467 ± 0.096 | 0.09 | -3.51 | 0.001052 |
| DMN-b PFCd 4 | 0.558 ± 0.066 | 0.488 ± 0.101 | 0.09 | -3.46 | 0.001215 |
| DMN-a PCC 1 | 0.573 ± 0.057 | 0.512 ± 0.069 | 0.07 | -3.46 | 0.001222 |
| DMN-a PFCm 3 | 0.507 ± 0.086 | 0.452 ± 0.075 | 0.08 | -3.43 | 0.001335 |
| SalVentAttn A ParMed7 | 0.497 ± 0.067 | 0.437 ± 0.057 | 0.06 | -3.34 | 0.001735 |

* *p*-values are uncorrected 2-sided

simple statistical approach we used, the sample size is simply too small (see limitations section). Separately, or in combination, these factors may have constrained our ability to detect a relationship between structural and functional damage in MCI.

The whole-brain graph theory measures revealed that the areas of difference between the two groups were centred on DMN nodes and the hippocampus. The analysis of the cluster coefficient (how well specialist information is segregated) showed that across both hemispheres there were fewer connections-between-nearest-neighbours of select DMN-a and DMN-b nodes and of the hippocampus in the MCI group, at cost levels of 0.10 and 0.15. This metric, related to the resilience of local networks, suggests that these areas in the MCI group are relatively more exposed to insult [114,115]. The presence of the hippocampus and DMN-a in the results from both analyses (functional connectivity strength and graph theory) may indicate that it has both lost connections and that its remaining connections are also weaker.

## Limitations

The heterogeneity of the MCI sample is a limiting factor in the interpretation of each analysis approach. The most common cause of MCI is AD-rleated pathology, however, other causes such as Lewy body disease and vascular insults in isolation or in combination are also common [6,7]. Further, not all cases of MCI go on to express further decline [116]. This heterogeneity, likely present in the current sample, introduces variation in the data that may hide or accentuate AD-related degeneration. Future studies incorporating protein-based diagnostic criteria will eventually minimise this confound. The current sample size is small and the power to detect a medium effect size (Cohen's d = 0.05) is approximately 52% in directionally predicted tests and 40% in two-sided tests (calculated using the pwr package for R [117]).

The rsMRI ROI approach is heavily dependent on the spatial accuracy of the boundaries of the chosen templates to reflect the functional organisation of the brain [118]. The difficulty in achieving perfect registration to such templates, particularly in the case of neurodegeneration, should be taken into consideration when interpreting results. Additionally, a region-of-interest approach, by definition, excludes brain regions from assessment and thus over-simplifies findings–in this case the differences between MCI and HC groups [34]. Also relating to the rsMRI data, bivariate correlations between regions were examined, this approach runs the risk of detecting spurious (or accentuating) connections between two areas if both those areas are connected to a common third area [118]. The group contrast of the current analysis may both help (by cancelling out common indirect connections) and hinder (by exposing spurious/indirect connections through contrast) this problem.

Constraints regarding the interpretation of graph theory analysis include those mentioned above for rsMRI (excluding the region of interest approach) and are extended by its binarisation process [119]. In order to achieve a high-level overview, it meant that in this instance, only a restricted range of cost % of connections based on correlation strength were included in the analysis. With this reductive approach valuable information is lost and the risk of a skewed understanding of clinically important brain connectivity differences is increased [118]. It should also be noted that a widespread difference in underlying functional connectivity between patient and control groups may introduce potential artifactual differences in network topology metrics [120].

Finally, a limitation of the DWI approach employed is the subjectivity introduced by the manual identification of ROIs and any cleaning of spurious tracts. To minimise this subjectivity we used published guides for the placement of tract delineators and the atlas-based approach whereby the ROI definition was applied across the entire group. We also facilitated natural variability by creating three atlases (small, medium, large) according to a subjective assessment of ventricle size. Despite these precautions we cannot eliminate this limitation, however, we can be confident that the results are directionally consistent with existing literature.

## Conclusion

We found white matter damage to the fornix and sub-divisions of left cingulum bundle, reduced connectivity strength within DMN-a, and reduced connectivity between the hippocampus and DMN-c, the hippocampus and DMN-a, and reduced information segregation (cluster coefficient) within the DMN and hippocampus in a group of MCI participants. However, we found no relationship between white matter disturbance and functional connectivity strength. This may be a reflection of independent degeneration processes in white and grey matter, particularly during early stage AD. Alternatively, the lack of relationship between the functional and structural measures may be related to study design and analytical factors.

## Supporting information

**S1 File.**
(DOCX)

## Acknowledgments

We thank Mr. Sojo Josephs for his assistance in acquiring the MRI data. We thank Cathy McHale, Joshi Dookhy, and members of the Memory Service at Tallaght University Hospital

and staff at St Patrick's University Hospital for their assistance in recruitment. We thank Jonathon McNulty for feedback on the manuscript.

## Author Contributions

**Conceptualization:** Seán P. Kennelly, Arun L. W. Bokde.

**Formal analysis:** Therese M. Gilligan, Francesca Sibilia.

**Funding acquisition:** Seán P. Kennelly, Arun L. W. Bokde.

**Investigation:** Therese M. Gilligan, Dervla Farrell.

**Methodology:** Seán P. Kennelly, Arun L. W. Bokde.

**Resources:** Declan Lyons, Seán P. Kennelly, Arun L. W. Bokde.

**Supervision:** Arun L. W. Bokde.

**Visualization:** Therese M. Gilligan.

**Writing – original draft:** Therese M. Gilligan.

**Writing – review & editing:** Therese M. Gilligan, Francesca Sibilia, Seán P. Kennelly, Arun L. W. Bokde.

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
