## [Decision Letter · Decision Letter 0]

8 Jul 2019

PONE-D-19-16445

No relationship between fornix and cingulum degradation and within-network decreases in functional connectivity in preclinical Alzheimer’s disease

PLOS ONE

Dear Dr Gilligan,

Thank you for submitting your manuscript to PLOS ONE. After careful consideration, we feel that it has merit but does not fully meet PLOS ONE’s publication criteria as it currently stands. Therefore, we invite you to submit a revised version of the manuscript that addresses the points raised during the review process.

The limited sample size could pose an issue questioning if the result is false negative (about the relationship between changes in structural connectivity and functional connectivity), as raised by the reviewer #2. The reviewer #1 also raised an important question about the network sparsity level and its potential influence to results. Please address them carefully in addition to other reviewers' comments.

We would appreciate receiving your revised manuscript by Aug 22 2019 11:59PM. To enhance the reproducibility of your results, we recommend that if applicable you deposit your laboratory protocols in protocols.io, where a protocol can be assigned its own identifier (DOI) such that it can be cited independently in the future. For instructions see: http://journals.plos.org/plosone/s/submission-guidelines#loc-laboratory-protocols

We look forward to receiving your revised manuscript.

Kind regards,

Han Zhang, Ph.D

Academic Editor

PLOS ONE

Journal Requirements:

2. Our internal editors have looked over your manuscript and determined that it may be within the scope of our Early Diagnosis and Treatment of Alzheimer's Disease Call for Papers. This collection of papers is headed by a team of Guest Editors for PLOS ONE: Michael Weiner, Roberta Brinton, Jussi Tohka and Yona Levites. With this Collection we hope to bring together researchers working on a wide range of disciplines, from molecular and preclinical work, through to patient-centered studies, including clinical trials.   Additional information can be found on our announcement page: https://collections.plos.org/s/alzheimersdisease. If you would like your manuscript to be considered for this collection, please let us know in your cover letter and we will ensure that your paper is treated as if you were responding to this call. Agreeing to part of the call-for-papers will not affect the date your manuscript is published. If you would prefer to remove your manuscript from collection consideration, please specify this in the cover letter.

Reviewers' comments:

Reviewer's Responses to Questions

**Comments to the Author**

1. Is the manuscript technically sound, and do the data support the conclusions?

Reviewer #1: Yes

Reviewer #2: Partly

2. Has the statistical analysis been performed appropriately and rigorously? 

Reviewer #1: Yes

Reviewer #2: I Don't Know

3. Have the authors made all data underlying the findings in their manuscript fully available?

Reviewer #1: Yes

Reviewer #2: Yes

4. Is the manuscript presented in an intelligible fashion and written in standard English?

Reviewer #1: Yes

Reviewer #2: Yes

5. Review Comments to the Author

Reviewer #1: This paper investigated changes in functional and structural connectivity among regions that support memory function in preclinical Alzheimer’s disease. Then investigated the relationship between white matter tracts (fornix, cingulum bundles, and uncinate fasciculus) and functional connectivity. It’s very interesting and meaningful. But there are some issues to be resolved as following.

Main concerns:

1. L313: “Cost (sparsity) was set at 0.15”. How about other costs? Are the results similar or seriously different?

2. L292: “Within hemisphere only analyses were conducted, across hemisphere connectivity was not inspected due to sample size”. It’s better and possible to make a comparison across hemisphere because authors didn’t use multivariate.

3. L272: “Its lmrob function fits a model based on an M-estimator using iteratively reweighted least squares estimation”. Could you list the model equation in details?

Minor concerns:

1. the figures aren’t with enough resolution, especially the title and number are unclear.

2. Table 3,4, why t(43), what does 43 mean? And the table format should be revised for consistency.

3. Some expressions are unclear, e.g.

L54: “……. more so than local neuropathological processes”.

L91: “ and to a lesser extent in MCI, found using a graph theory approach ”

Reviewer #2: The study used multimodal MRI data including DWI and resting state MRI to compared the difference between MCI and AD. This study has some strengths, such as the advanced analytical method. But there are still several important details need to be clarified.

1. The concept of preclinical stage of AD is definitely not MCI. Cited from a review (Epelbaum et al., 2017), three main issues concerning the concept of preclinical AD must be clarified: 1) the level of cognitive performance considered as normal cognition, 2) the changes in cognitive performance considered as cognitive decline, and 3) the best biomarkers or the best combination of them able to identify the “AD pathophysiological signature” in vivo. Hence, the stage of MCI should be prodromal AD, but not preclinical. Please clarify all related content.

Ref: Epelbaum S, Genthon R, Cavedo E, et al. Preclinical Alzheimer's disease: A systematic review of the cohorts underlying the concept [J]. Alzheimers Dement, 2017, 13(4): 454-67.

2. In the section of Introduction, I think it would have been better to propose the research aims at the last paragraph after review all the related studies and their limitation.

3. In line 142, the cut-off score in GDS is 5?

4. In Line 275-277,” We chose not to include depression as a control covariate given that its presence may reflect dementia pathology rather than a reaction to memory problems.” The relationship between depression and cognitive function is very complicated, part of the depressive elderly would never convert to dementia. This sentence is confusing. Please clarify and revise related analysis result.

5. The author should clarify the definition of DMN-a, -b and -c, because it seems very confusing for a clinician reading.

6. Due to the small sample size, I think it is better to compare with age, gender matched controls. In addition, the age distribution was “54-80”. For neuroimage study, especially AD-related study, age is a very important factor.

7. About method, there are some significant difference between group in neuropsychological test, such as MMSE, word delay test, so what’s the relationship between the neuroimage biomarker with these clinical data. Please do some further correlation analysis which may reflect the clinical implication.

6. PLOS authors have the option to publish the peer review history of their article (what does this mean?). If published, this will include your full peer review and any attached files.

Reviewer #1: No

Reviewer #2: Yes: Ling Yue

---

## [Author Response · Author response to Decision Letter 0]

21 Aug 2019

Reviewer #1: 

This paper investigated changes in functional and structural connectivity among regions that support memory function in preclinical Alzheimer’s disease. Then investigated the relationship between white matter tracts (fornix, cingulum bundles, and uncinate fasciculus) and functional connectivity. It’s very interesting and meaningful. But there are some issues to be resolved as following.

Overall Reply: We thank the reviewer for their comments, and their appreciation and help in producing the improvements made to the manuscript. 

Main concerns:

1. L313: “Cost (sparsity) was set at 0.15”. How about other costs? Are the results similar or seriously different?

Thank you for this feedback. Given our start of 0.15 we chose to investigate this further using the additional levels: 0.10, 0.20, and 0.25. 

We have inserted the following text within the results section:

Given the challenges of defining a-priori an appropriate network cost level (94), and following feedback, we additionally inspected the graph theory measures at sparsity levels of .10, .20 and .25. There were significant differences between the two groups in the cluster coefficient measure for sparsity levels of 0.10, 0.20 and 0.25 (uncorrected for multiple comparisons across levels). As per cost level 0.15, the cluster coefficient measures were higher in the HC compared to the MCI group. At level 0.10 the difference was driven by 16 ROIs (12 from the DMN and hippocampus – 10 of which drive the group differences at sparsity level 0.15). At level 0.20 no individual ROI survived correction for multiple comparisons. At level 0.25 the difference was driven exclusively by LH temporal area 3 of the DMN_b (it was also a driver of differences at levels 0.10 and 0.15). Full details are provided in S1 file. There was a significant difference in the average path length between the groups at sparsity level of 0.10. As per sparsity level 0.15, this average path length difference was not present at levels 0.20 and 0.25. Further details are provided in S1 file.

We have added the below highlighted piece of text in the results section:

The whole-brain graph theory measures revealed that the areas of difference between the two groups were centred on DMN nodes and the hippocampus. The analysis of the cluster coefficient (how well specialist information is segregated) showed that across both hemispheres there were fewer connections-between-nearest-neighbours of select DMN-a and DMN-b nodes and of the hippocampus in the MCI group, at cost levels of 0.10 and 0.15. This metric, related to the resilience of local networks, suggests that these areas in the MCI group are relatively more exposed to insult (114,115). The presence of the hippocampus and DMN-a in the results from both analyses (functional connectivity strength and graph theory) may indicate that it has both lost connections and that its remaining connections are also weaker. 

We have added the following highlighted pieces of text to the Limitations section:

Constraints regarding the interpretation of graph theory analysis include those mentioned above for rsMRI (excluding the region of interest approach) and are extended by its binarisation process (119). In order to achieve a high-level overview, it meant that in this instance, only a restricted range of cost % of connections based on correlation strength were included in the analysis. With this reductive approach valuable information is lost and the risk of a skewed understanding of clinically important brain connectivity differences is increased (118). It should also be noted that a widespread difference in underlying functional connectivity between patient and control groups may introduce potential artifactual differences in network topology metrics (120).

We have inserted the following details within the supplementary S1 file:

Table H: Graph theory cluster coefficient, sparsity level 0.10

Measure HC MCI Estimate Statistic P-value*

LH 

DMN-b Temp 3 0.558 ± 0.058 0.460 ± 0.100 0.10 4.25 0.000114

DMN-a PCC 1 0.575 ± 0.079 0.490 ± 0.082 0.10 4.14 0.000161

DMN-a PCC 2 0.555 ± 0.054 0.496 ± 0.075 0.07 3.70 0.000608

SomMot A 16 0.531 ± 0.0 0.423 ± 0.094 0.11 3.60 0.000808

DMN-b PFCd 1 0.530 ± 0.071 0.453 ± 0.084 0.08 3.58 0.000871

Hippocampus 0.498 ± 0.126 0.369 ± 0.132 0.14 3.49 0.001137

DMN-a PFCm 3 0.561 ± 0.077 0.483 ± 0.094 0.09 3.47 0.001198

DMN-b Temp 2 0.568 ± 0.089 0.478 ± 0.097 0.09 3.33 0.001781

RH 

Hippocampus 0.527 ± 0.125 0.357 ± 0.136 0.20 5.53 0.000002

DMN-a PFCm 1 0.536 ± 0.068 0.450 ± 0.080 0.09 4.12 0.000170

DMN-b PFCd 4 0.555 ± 0.076 0.475 ± 0.107 0.10 3.72 0.000572

Limibic-a OFC 6 0.553 ± 0.079 0.453 ± 0.112 0.11 3.60 0.000818

SalVentAttn A ParMed 8 0.500 ± 0.101 0.426 ± 0.102 0.10 3.49 0.001124

SalVentAttn A ParMed 4 0.487 ± 0.094 0.399 ± 0.090 0.10 3.46 0.001235

DMN-a PFCd 2 0.555 ± 0.078 0.463 ± 0.095 0.09 3.45 0.001268

DMN-a PCC 1 0.559 ± 0.064 0.494 ± 0.077 0.07 3.36 0.001649

* p-values are uncorrected 2-sided

Graph theory analysis, at sparsity level 0.10, revealed a significant between-group difference in the cluster coefficient measure. This was higher in the HC (M=0.484, SD=0.027) compared to the MCI (M=0.461, SD=0.035) group (b = .03, t = 2.90, puncor = .006). See table H above.

Graph theory analysis, at sparsity level 0.20, revealed a significant between-group difference in the cluster coefficient measure. This was higher in the HC (M=0.504, SD=0.025) compared to the MCI (M=0.490, SD=0.031) group (b = .02, t = 2.31, puncor = .026). The ROIs driving this difference did not survive correction for multiple comparisons.

Graph theory analysis, at sparsity level 0.25, revealed a significant between-group difference in the cluster coefficient measure. This was higher in the HC (M=0.521, SD=0.024) compared to the MCI (M=0.509, SD=0.029) group (b = .02, t = 2.19, puncor = .034). See table I.

Table I: Graph theory cluster coefficient, sparsity level 0.25

Measure HC MCI Estimate Statistic P-value*

LH 

DMN-b Temp 3 0.580 ± 0.051 0.512 ± 0.070 0.08 4.69 0.00028

* p-values are uncorrected 2-sided

Graph theory analysis, at sparsity level 0.10, revealed a statistical difference in average path length between the groups, with a longer path length in HC compared to controls (b = .05, t = 2.42, puncor = .020; HC: M = 2.39, SD = 0.071; MCI: M = 2.35, SD = 0.084). This difference was driven by two ROIs in the somatomotor network - one in the left hemisphere and one in the right.

Table J: Graph theory average path length, sparsity level 0.10

Measure HC MCI Estimate Statistic P-value*

LH 

SomMot-a 15 2.46 ± 0.215 2.22. ± 0.180 0.25 4.25 0.000112

RH 

SomMot-a 13 2.45 ± 0.129 2.25 ± 0.200 0.19 4.20 0.000133

* p-values are uncorrected 2-sided

Graph theory analysis did not reveal a stastical difference in average path length between the two groups at sparsity level 0.20: (b = .02, t = 1.54, puncor = .130; HC: M = 1.89, SD = 0.035; MCI: M = 1.88, SD = 0.040), nor at sparsity level 0.25: (b = .01, t = 1.11, puncor = .275; HC: M = 1.78, SD = 0.024; MCI: M = 1.78, SD = 0.029).

2. L292: “Within hemisphere only analyses were conducted, across hemisphere connectivity was not inspected due to sample size”. It’s better and possible to make a comparison across hemisphere because authors didn’t use multivariate.

We agree with the reviewer that such comparisons of functional connectivity are of interest. However, we were confronted with the statistical limitations of our sample size. With seven regions of interest per hemisphere, it would require 91 comparisons to inspect all possible pairings within and across hemispheres. This, alongside the requisite correction for multiple comparisons, was deemed a-priori to be beyond the capacity of this sample’s data.

It is also worth highlighting that analysis of the limbic white matter revealed differences between the hemispheres. However, because the focus was on limbic white matter tracts we did not analyse the corpus callosum. The aforementioned statistical considerations aside, these points taken together, would complicate any interpretation of across hemisphere functional relationships (if we had found them).

3. L272: “Its lmrob function fits a model based on an M-estimator using iteratively reweighted least squares estimation”. Could you list the model equation in details?

Here is an indicative example of how the formula was run in R:

lmrob(Mean_FA ~ Group + Age + Gender + Education, data = L_)

This approach was chosen because it is more appropriate for small sample sizes and the attendant problems of meeting test assumptions, including the issue of having a large number of parameters or covariates to model relative to the sample size, as per this study. This approach was chosen based on the recommendation of Field and Wilcox (2017): 

“M-estimators determine whether a score is an outlier empirically and if it is, adjustments are made for it. The adjustment could be to completely ignore the observation or to down-weight it. Obvious advantages of M-estimators are that you can (1) down-weight rather than exclude observations; (2) avoid over- or under-trimming your data; and (3) perform non-symmetric trimming.” (p 23, 1)

1. Field AP, Wilcox RR. Robust statistical methods: A primer for clinical psychology and experimental psychopathology researchers. Behav Res Ther [Internet]. 2017;98:19–38. Available from: http://dx.doi.org/10.1016/j.brat.2017.05.013

“This function computes an MM-type regression estimator as described in Yohai (1987) and Koller and Stahel (2011). By default it uses a bi-square redescending score function, and it returns a highly robust and highly efficient estimator (with 50% breakdown point and 95% asymptotic efficiency for normal errors).” (p 67, 2)

2. Maecheler M, Rousseeuw P, Croux C, Todorov V, Rucksuhl A, Salibian-Barrera M, et al. “robustbase”: Basic robust statistics. R package [Internet]. 2018. Available from: http://cran.r-project.org/package=robustbase

A number of formulae are required to calculate the robust regression. With due deference to the field of the statistics, we refer the reviewer to the source materials for a fuller appreciation of the approach:

3. Koller M, Stahel WA. Sharpening Wald-type inference in robust regression for small samples. Comput Stat Data Anal [Internet]. 2011;55(8):2504–15. Available from: http://dx.doi.org/10.1016/j.csda.2011.02.014

4. Yohai, V., 1987. High breakdown-point and high efficiency robust estimates for regression. The Annals of Statistics 15 (2), 642–656.

Minor concerns:

1. the figures aren’t with enough resolution, especially the title and number are unclear.

Thank you for highlighting this frustration. We followed the guidelines provided and used the recommended PACE website to ensure the figures met publication standards. Should the paper be accepted, perhaps this is something we can work with the publication team to resolve?

2. Table 3,4, why t(43), what does 43 mean? And the table format should be revised for consistency.

t(43) refers to the t-statistic for 43 degrees of freedom (allowing for 2 groups, with age, gender and education as control covariates). As suggested, we have changed the text to ‘statistic’ so that it is consistent with the other tables.

3. Some expressions are unclear, e.g.

L54: “……. more so than local neuropathological processes”.

We have revised the sentence to improve clarity:

“For example, it is proposed that early hypometabolism seen in the posterior cingulate cortex reflects distant damage in the hippocampal formation more so than local neuropathological processes within the posterior cingulate cortex (9,10).”

L91: “ and to a lesser extent in MCI, found using a graph theory approach ”

We have revised the sentence to improve clarity:

“That finding of increased functional connectivity parallels the lack of segregation between the DMN and frontoparietal networks that have been revealed using a graph theory approach in AD, and to a lesser extent, MCI patients.”

Reviewer #2: 

The study used multimodal MRI data including DWI and resting state MRI to compared the difference between MCI and AD. This study has some strengths, such as the advanced analytical method. But there are still several important details need to be clarified.

Overall Reply: We thank Dr Yue for their comments, and their appreciation and help in producing the improvements made to the manuscript. 

For the sake of avoiding confusion in the below responses (in case the paper is not to hand), we wish to clarify that the study compared differences between a group with MCI and a matched healthy control group.

1. The concept of preclinical stage of AD is definitely not MCI. Cited from a review (Epelbaum et al., 2017), three main issues concerning the concept of preclinical AD must be clarified: 1) the level of cognitive performance considered as normal cognition, 2) the changes in cognitive performance considered as cognitive decline, and 3) the best biomarkers or the best combination of them able to identify the “AD pathophysiological signature” in vivo. Hence, the stage of MCI should be prodromal AD, but not preclinical. Please clarify all related content.

Ref: Epelbaum S, Genthon R, Cavedo E, et al. Preclinical Alzheimer's disease: A systematic review of the cohorts underlying the concept [J]. Alzheimers Dement, 2017, 13(4): 454-67.

Thank you for pointing this out and for providing the relevant reference. We have removed the two references to preclinical AD, in the title and in the abstract (line 15), and replaced with prodromal AD:

Title: “No relationship between fornix and cingulum degradation and within-network decreases in functional connectivity in prodromal Alzheimer’s disease”

Line 15: “We investigated changes in functional and structural connectivity among regions that support memory function in prodromal Alzheimer’s disease, i.e., during the mild cognitive impairment (MCI) stage.”

2. In the section of Introduction, I think it would have been better to propose the research aims at the last paragraph after review all the related studies and their limitation.

We agree that is helpful to place the purpose of the study at the end of the introduction section. We had previously received feedback from colleagues urging us to provide context in the initial paragraph of the introduction. It seems to be an important point for the reader. For these reasons we attempted to address both viewpoints by mentioning aims in both the first and last paragraph of the introduction:

First paragraph of introduction:

“This study aims to probe the relationship between structural and functional brain changes in MCI through use of resting state magnetic resonance imaging (rsMRI) and diffusion weighted imaging (DWI).”

Last paragraph of introduction:

“This study adds to the literature by investigating constrained spherical deconvolution DWI measures (54) of MCI-targeted white matter tracts (fornix, cingulum bundles, and uncinate fasciculus) and temporal correlation connectivity measures of MCI-targeted functional networks (DMN and limbic), and by examining the relationship between those structural and functional measures.”

Admittedly this approach is unlikely to be agreeable to all stylistic perspectives. However, we hope it does not deter from the overall readability and understanding of the paper and its aims.

3. In line 142, the cut-off score in GDS is 5?

Thank you for picking this up. We used the short-form of the GDS, it has a suggested cut-off of 5. We have inserted ‘short-form’ at the first mention of the use of the GDS in our study, in line 151:

‘……short-form Geriatric Depression Scale (GDS; (59))’

4. In Line 275-277,” We chose not to include depression as a control covariate given that its presence may reflect dementia pathology rather than a reaction to memory problems.” The relationship between depression and cognitive function is very complicated, part of the depressive elderly would never convert to dementia. This sentence is confusing. Please clarify and revise related analysis result.

Thank you for bringing our attention to this section of text. We have inserted the below text within the analysis section. The text now emphasizes the complication of depression and provides more clarity around the reasons for our analysis choice. Now line 299-304:

The relationship between depression and cognitive function is complicated and some depressive elderly will not convert to dementia (84). However, we chose not to include depression as a control covariate given that its presence may reflect dementia pathology (85). That is, controlling for depression would run the risk of removing relevant explanatory variance. Further, in this group it would both reduce sample size as one participant declined to complete the GDS and risk over-fitting the model. In any case, in this cohort depression did not correlate with worsening measures of cognition (see Fig B in S1 file).

Further, we have clarified the details on missing neuropsychological information in the methods section:

“Five MCI people declined to complete the entirety of the neuropsychological testing set (four did not complete the CERAD tests, one did not complete the CERAD or GDS tests) – however, all were successfully scanned.”

5. The author should clarify the definition of DMN-a, -b and -c, because it seems very confusing for a clinician reading.

Thank you for sharing this perspective. The atlas we used (Schaefer et al, 2018) features a relatively recent and specific computational-based segmentation of the default mode network based on functional connectivity. 

These parcellations were defined using task and resting state fMRI data of 1489 young adults (18-35 years old) and a computational method that made the optimal combination of two different segmentation approaches: 1) abrupt changes in functional connectivity patterns such as might be defined histologically and 2) parcels of similar functional activity identified independently of spatial considerations. This parcellation method is thought to reflect functional brain organization in a neurobiologically meaningful way. 

Alongside creating functional anatomical clusters, the parcellations are helpful from a research and statistical perspective (it moves on from examining every single voxel in a MRI dataset). In order to address different research needs, the authors provided parcellations in varying amounts of clusters (400, 600, 800, 1000), i.e., from diffuse to focal levels. In our study we chose the atlas with 400 parcellations. In that instance it revealed a default mode network that could be meaningfully sub-divided into 3 functional clusters, and a limbic network that could be divided into two functional clusters. Within each of these clusters focal sources of the functional activity can be further labelled. In figure 2 of the paper, we provided a view of the focal sources within each cluster of the default mode and limbic networks.

We have amended the text as follows:

Using the 2mm 400 region cortical atlas (80) two limbic (a and b) and three default mode sub-networks (a, b, and c) per hemisphere were identified. These atlas parcellations were computed from functional connectivity patterns. The sources of the functional connectivity signals within each sub-network are detailed in Fig 2. Centroid co-ordinates for the parcels are presented in S1 file table L.

Due to its length, table L has been placed at the end of the present document.

Schaefer A, Kong R, Gordon EM, Laumann TO, Zuo X, Holmes AJ, et al. Local-Global Parcellation of the Human Cerebral Cortex from Intrinsic Functional Connectivity MRI. Cereb Cortex. 2018;28:3095–114.

6. Due to the small sample size, I think it is better to compare with age, gender matched controls. In addition, the age distribution was “54-80”. For neuroimage study, especially AD-related study, age is a very important factor. 

We agree with the reviewer that age is an important consideration. While the healthy control group is unfortunately not a precise like-for-like of the MCI group, there were no statistically significant demographic differences between the two groups. We also took the step of including age, gender and education as control covariates in the linear regressions.

Taking the opportunity to reflect on the reviewer’s comment regarding the sample more generally, we have included an additional sentence (L504) to highlight the possibility that the lack of finding of a relationship between functional and structural connectivity is due to the sample size constraints:

“It is also possible that, notwithstanding the simple statistical approach we used, the sample size is simply too small (see limitations section).”

7. About method, there are some significant difference between group in neuropsychological test, such as MMSE, word delay test, so what’s the relationship between the neuroimage biomarker with these clinical data. Please do some further correlation analysis which may reflect the clinical implication.

Thank you for raising this point, we do appreciate that such an analysis could hold interest. We chose not to explore this aspect primarily because it falls outside the focus of the paper. It also risks ending up reporting spurious relations due to statistical limitations. There are a couple of reasons for this. First, there is the potential to investigate multiple relationships – there are 50 metrics for the DWI data and 52 for the functional connectivity data that could each be investigated for relationships with up to 11 neuropsychological measures. Even limiting the analysis to a post-hoc exploration between those imaging and neuropsychological measures that were significantly different between groups would still require 120 tests. Second, as mentioned above, five MCI patients have incomplete neuropsychological test sets thus reducing the sample size available for such correlations to twenty patients. For these reasons, albeit that we appreciate the rationale for the suggestion, we would prefer not to undertake those analyses. The data is being made publicly available and we sincerely hope that it might be of use to other researchers who may be in a position to combine it with other datasets to create the statistical bandwidth to answer such questions.

Reviewer #2, point 5: Additional information for the supplementary file:

Table L: Schaefer atlas 2018 400 parcels 17 networks MNI152 2mm – centroid co-ordinates for default mode and limbic networks. Elements shaded in grey were implicated in one/more graph theory analyses. For complete list of centroids see: https://github.com/ThomasYeoLab/CBIG/blob/master/stable_projects/brain_parcellation/Schaefer2018_LocalGlobal/Parcellations/MNI/Centroid_coordinates/Schaefer2018_400Parcels_17Networks_order_FSLMNI152_2mm.Centroid_RAS.csv

ROI Index Label Name X Y Z

36 17Networks_LH_SomMotA_12 -24 -10 64

39 17Networks_LH_SomMotA_15 -4 -26 68

40 17Networks_LH_SomMotA_16 -14 -12 72

109 17Networks_LH_Limbic_OFC_1 -12 24 -20

110 17Networks_LH_Limbic_OFC_2 -24 22 -20

111 17Networks_LH_Limbic_OFC_3 -10 48 -22

112 17Networks_LH_Limbic_OFC_4 -4 24 -20

113 17Networks_LH_Limbic_OFC_5 -16 64 -8

114 17Networks_LH_Limbic_TempPole_1 -38 -6 -42

115 17Networks_LH_Limbic_TempPole_2 -24 6 -40

116 17Networks_LH_Limbic_TempPole_3 -26 -10 -32

117 17Networks_LH_Limbic_TempPole_4 -54 -22 -30

118 17Networks_LH_Limbic_TempPole_5 -40 -22 -26

119 17Networks_LH_Limbic_TempPole_6 -32 12 -30

120 17Networks_LH_Limbic_TempPole_7 -44 6 -16

149 17Networks_LH_DefaultA_IPL_1 -48 -64 32

150 17Networks_LH_DefaultA_IPL_2 -42 -72 44

151 17Networks_LH_DefaultA_PFCd_1 -24 28 44

152 17Networks_LH_DefaultA_PFCd_2 -18 36 48

153 17Networks_LH_DefaultA_PFCd_3 -22 20 52

154 17Networks_LH_DefaultA_PCC_1 -4 -54 20

155 17Networks_LH_DefaultA_PCC_2 -6 -60 30

156 17Networks_LH_DefaultA_PCC_3 -8 -44 32

157 17Networks_LH_DefaultA_PCC_4 -4 -34 38

158 17Networks_LH_DefaultA_PCC_5 -2 -16 38

159 17Networks_LH_DefaultA_PCC_6 -2 -68 42

160 17Networks_LH_DefaultA_PCC_7 -6 -50 42

161 17Networks_LH_DefaultA_PFCm_1 -4 56 -10

162 17Networks_LH_DefaultA_PFCm_2 -6 36 -8

163 17Networks_LH_DefaultA_PFCm_3 -6 60 6

164 17Networks_LH_DefaultA_PFCm_4 -6 44 6

165 17Networks_LH_DefaultA_PFCm_5 -16 68 8

166 17Networks_LH_DefaultA_PFCm_6 -6 34 20

167 17Networks_LH_DefaultB_Temp_1 -44 12 -34

168 17Networks_LH_DefaultB_Temp_2 -54 -2 -30

169 17Networks_LH_DefaultB_Temp_3 -62 -18 -20

170 17Networks_LH_DefaultB_Temp_4 -56 -8 -14

171 17Networks_LH_DefaultB_Temp_5 -60 -34 -4

172 17Networks_LH_DefaultB_Temp_6 -52 -22 -6

173 17Networks_LH_DefaultB_IPL_1 -46 -58 20

174 17Networks_LH_DefaultB_IPL_2 -56 -54 30

175 17Networks_LH_DefaultB_PFCd_1 -4 52 28

176 17Networks_LH_DefaultB_PFCd_2 -14 58 30

177 17Networks_LH_DefaultB_PFCd_3 -22 50 32

178 17Networks_LH_DefaultB_PFCd_4 -8 42 52

179 17Networks_LH_DefaultB_PFCd_5 -12 24 60

180 17Networks_LH_DefaultB_PFCd_6 -6 10 64

181 17Networks_LH_DefaultB_PFCl_1 -40 20 48

182 17Networks_LH_DefaultB_PFCl_2 -42 8 48

183 17Networks_LH_DefaultB_PFCv_1 -36 22 -16

184 17Networks_LH_DefaultB_PFCv_2 -36 36 -12

185 17Networks_LH_DefaultB_PFCv_3 -46 32 -10

186 17Networks_LH_DefaultB_PFCv_4 -48 28 0

187 17Networks_LH_DefaultB_PFCv_5 -54 20 12

188 17Networks_LH_DefaultC_IPL_1 -40 -78 30

189 17Networks_LH_DefaultC_Rsp_1 -14 -48 4

190 17Networks_LH_DefaultC_Rsp_2 -8 -52 10

191 17Networks_LH_DefaultC_Rsp_3 -14 -60 18

192 17Networks_LH_DefaultC_PHC_1 -20 -20 -26

193 17Networks_LH_DefaultC_PHC_2 -30 -32 -18

194 17Networks_LH_DefaultC_PHC_3 -18 -38 -12

236 17Networks_RH_SomMotA_13 10 -40 68

299 17Networks_RH_SalVentAttnA_ParMed_4 6 10 58

302 17Networks_RH_SalVentAttnA_ParMed_7 6 -2 66

303 17Networks_RH_SalVentAttnA_ParMed_8 16 6 70

313 17Networks_RH_Limbic_OFC_1 14 24 -20

314 17Networks_RH_Limbic_OFC_2 22 22 -20

315 17Networks_RH_Limbic_OFC_3 8 46 -24

316 17Networks_RH_Limbic_OFC_4 20 42 -18

317 17Networks_RH_Limbic_OFC_5 4 22 -20

318 17Networks_RH_Limbic_OFC_6 10 62 -14

319 17Networks_RH_Limbic_TempPole_1 28 -2 -40

320 17Networks_RH_Limbic_TempPole_2 48 -6 -40

321 17Networks_RH_Limbic_TempPole_3 36 18 -38

322 17Networks_RH_Limbic_TempPole_4 40 -14 -32

323 17Networks_RH_Limbic_TempPole_5 28 12 -30

324 17Networks_RH_Limbic_TempPole_6 50 -28 -26

358 17Networks_RH_DefaultA_Temp_1 60 -8 -24

359 17Networks_RH_DefaultA_IPL_1 54 -54 26

360 17Networks_RH_DefaultA_IPL_2 48 -64 42

361 17Networks_RH_DefaultA_PFCd_1 26 34 38

362 17Networks_RH_DefaultA_PFCd_2 24 26 50

363 17Networks_RH_DefaultA_PCC_1 6 -52 24

364 17Networks_RH_DefaultA_PCC_2 4 -64 32

365 17Networks_RH_DefaultA_PCC_3 6 -38 34

366 17Networks_RH_DefaultA_PCC_4 4 -20 36

367 17Networks_RH_DefaultA_PCC_5 10 -52 36

368 17Networks_RH_DefaultA_PFCm_1 6 42 -10

369 17Networks_RH_DefaultA_PFCm_2 10 66 0

370 17Networks_RH_DefaultA_PFCm_3 8 42 4

371 17Networks_RH_DefaultA_PFCm_4 8 54 12

372 17Networks_RH_DefaultA_PFCm_5 18 64 16

373 17Networks_RH_DefaultA_PFCm_6 6 26 18

374 17Networks_RH_DefaultB_Temp_1 64 -24 -8

375 17Networks_RH_DefaultB_Temp_2 64 -38 0

376 17Networks_RH_DefaultB_AntTemp_1 50 8 -32

377 17Networks_RH_DefaultB_PFCd_1 6 58 30

378 17Networks_RH_DefaultB_PFCd_2 16 52 36

379 17Networks_RH_DefaultB_PFCd_3 4 44 40

380 17Networks_RH_DefaultB_PFCd_4 14 38 52

381 17Networks_RH_DefaultB_PFCd_5 12 20 62

382 17Networks_RH_DefaultB_PFCv_1 34 22 -18

383 17Networks_RH_DefaultB_PFCv_2 48 32 -8

384 17Networks_RH_DefaultB_PFCv_3 54 24 6

385 17Networks_RH_DefaultC_IPL_1 48 -64 22

386 17Networks_RH_DefaultC_IPL_2 46 -76 30

387 17Networks_RH_DefaultC_Rsp_1 14 -46 4

388 17Networks_RH_DefaultC_Rsp_2 12 -56 16

389 17Networks_RH_DefaultC_PHC_1 22 -18 -28

390 17Networks_RH_DefaultC_PHC_2 30 -30 -18

---

## [Decision Letter · Decision Letter 1]

5 Sep 2019

[EXSCINDED]

PONE-D-19-16445R1

No relationship between fornix and cingulum degradation and within-network decreases in functional connectivity in prodromal Alzheimer’s disease

PLOS ONE

Dear Dr Gilligan,

Thank you for submitting your manuscript to PLOS ONE. After careful consideration, we feel that it has merit but does not fully meet PLOS ONE’s publication criteria as it currently stands. Therefore, we invite you to submit a revised version of the manuscript that addresses the points raised during the review process. 

Specifically, please make clarification regarding reviewer #1's minor concern. It may not need a re-review. 

We would appreciate receiving your revised manuscript by Oct 20 2019 11:59PM. To enhance the reproducibility of your results, we recommend that if applicable you deposit your laboratory protocols in protocols.io, where a protocol can be assigned its own identifier (DOI) such that it can be cited independently in the future. For instructions see: http://journals.plos.org/plosone/s/submission-guidelines#loc-laboratory-protocols

We look forward to receiving your revised manuscript.

Kind regards,

Han Zhang, Ph.D

Academic Editor

PLOS ONE

Additional Editor Comments (if provided):

Please address the reviewer #1's minor concern.

Reviewers' comments:

Reviewer's Responses to Questions

**Comments to the Author**

1. If the authors have adequately addressed your comments raised in a previous round of review and you feel that this manuscript is now acceptable for publication, you may indicate that here to bypass the “Comments to the Author” section, enter your conflict of interest statement in the “Confidential to Editor” section, and submit your "Accept" recommendation.

Reviewer #1: (No Response)

Reviewer #2: All comments have been addressed

2. Is the manuscript technically sound, and do the data support the conclusions?

Reviewer #1: Yes

Reviewer #2: Yes

3. Has the statistical analysis been performed appropriately and rigorously? 

Reviewer #1: Yes

Reviewer #2: Yes

4. Have the authors made all data underlying the findings in their manuscript fully available?

Reviewer #1: Yes

Reviewer #2: Yes

5. Is the manuscript presented in an intelligible fashion and written in standard English?

Reviewer #1: Yes

Reviewer #2: Yes

6. Review Comments to the Author

Reviewer #1: Minor concerns:

The sign of static value in different tables may not be consistent. For example, in table 2, positive statistic value means increase in MCI subjects, negative means decrease. But in table 3, they show opposite meaning according to the results description.

The statistic value means T value? It's better to be clearer in each table.

Reviewer #2: The authors addressed all the comments adequately, and their manuscript is significantly improved. I have no further comments.

7. PLOS authors have the option to publish the peer review history of their article (what does this mean?). If published, this will include your full peer review and any attached files.

Reviewer #1: No

Reviewer #2: No

---

## [Author Response · Author response to Decision Letter 1]

11 Sep 2019

We thank both reviewers for again spending time and giving great attention to improving the paper. We are appreciative of your support.

Reviewer #1: Minor concerns:

The sign of static value in different tables may not be consistent. For example, in table 2, positive statistic value means increase in MCI subjects, negative means decrease. But in table 3, they show opposite meaning according to the results description.

The statistic value means T value? It's better to be clearer in each table.

Reply: You are correct. Differences in how the reference group was defined in the various analyses files are behind these inconsistencies. However, as you point out the explanation for said differences had been correctly described in the manuscript.

We have amended the text for the DWI results and its corresponding Table 2 in the main manuscript:

All diffusion measures of the fornix showed evidence of degeneration in the MCI group: MD, Da and Dr were comparatively increased in the left (MD: t = 4.21, pcor = .0006, Da: t = 5.32, pcor = .00005, Dr: t = 3.74, pcor = .002) and the right fornix (MD: t = 5.06, pcor < .0001, Da: t = 5.59, pcor = .00005, Dr: t = 4.65, pcor = .0002), and FA was comparatively decreased in the left (t = -2.78, pcor = .019) and right (t = -2.42, pcor = .035) fornices. In the MCI compared to the HC group, MD and Da were significantly increased in the left parahippocampal cingulum (MD: t = 2.72, pcor = .019, Da: t = 3.02, pcor = .012), and MD and Dr were significantly increased in the left retrosplenial cingulum (MD: t = 3.08, pcor = .011, Dr: t = 2.43, pcor = .035) and in the left subgenual cingulum (MD: t = 3.08, pcor = .011, Dr: t = 2.77, pcor = .019). 

Table 3. MD values per structure per hemisphere 

Structure Hemi HC MCI Estimate t-statistic p-value *

Fornix LH 0.00120 ± 0.00008 0.00132 ± 0.00014 0.000139 4.210 0.000064

 RH 0.00118 ± 0.00008 0.00127 ± 0.00008 0.000100 5.056 .0000042

Para-hippocampal cingulum LH 0.00072 ± 0.00003 0.00075 ± 0.00005 0.0000187 2.721 0.00467

 RH 0.00072 ± 0.00003 0.00074 ± 0.00005 0.0000104 1.007 0.15950

Retrosplenial cingulum LH 0.00067 ± 0.00002 0.00069 ± 0.00002 0.0000213 3.084 0.00178

 RH 0.00067 ± 0.00002 0.00068 ± 0.00003 0.0000148 2.052 0.02315

Subgenual cingulum LH 0.00069 ± 0.00002 0.00070 ± 0.00002 0.00001516 3.078 0.00181

 RH 0.00069 ± 0.00002 0.00069 ± 0.00002 0.00000946 1.706 0.04765

Uncinate fasciculus LH 0.00070 ± 0.00002 0.00071 ± 0.00004 0.00000444 0.806 0.21225

 RH 0.00072 ± 0.00002 0.00072 ± 0.00003 0.00000254 0.336 0.36900

Hemi = hemisphere, LH = left hemisphere, RH = right hemisphere. *p-values are uncorrected, 1-sided

We have corrected the text for the rsMRI results (tables were already correct):

In MCI compared to HC, statistically smaller within-network connectivity was found in the DMN-a in the left (t = -3.38, pcor = .010) and right (t = -3.75, pcor = .005) hemispheres. See Table 4.

In MCI compared to HC, statistically smaller between-network connectivity was found between DMN-a and DMN-c, between DMN-a and the hippocampus, and between DMN-c and the hippocampus in both the left hemisphere (respectively: t = -4.63, pcor = .002; t = -4.21, pcor = .003; t = -3.99, pcor = .003) and the right hemisphere (respectively: t = -3.81, pcor = .005; t = -3.31, pcor = .011; t = -3.52, pcor = .008). In the right hemisphere only in MCI compared to HC, statistically smaller between-network connectivity was found between DMN-c and the thalamus (t = -3.02, pcor = .021) and between DMN-c and the limbic-a network (t = -4.01, pcor = .003). See Table 5, and Figs 5 and 6, for results mentioned here and Table G in S1 file for the complete set.

We have amended text and table for the graph theory results:

Graph theory analyses, at sparsity level of 0.15, revealed a significant between-group difference in the cluster coefficient measure. This was higher in the HC (M=0.492, SD=0.027) compared to the MCI (M=0.474, SD=0.033) group (b = .02, t = -2.53, puncor = .015). This difference was driven by 18 ROIs - 8 in DMN-a (5 RH, 3 LH), 5 in DMN-b (2 RH, 3LH), both hippocampi, two ROIs in the LH somatomotor network and one in the RH salient ventral attention network – all of which survived FDR-correction. See Fig 7 and Table 6 for details. The difference in average path length between the two groups did not reach statistical significance (b = .03, t = -1.94, puncor = .059; HC: M=2.08, SD=0.049; MCI: M=2.06, SD=0.06).

Table 6. Graph theory cluster coefficient, sparsity level of 0.15

Measure HC MCI Estimate t-statistic P-value*

LH 

DMN-a PCC 2 0.569 ± 0.045 0.510 ± 0.067 0.07 -4.17 0.000146

DMN-b Temp 3 0.564 ± 0.054 0.481 ± 0.096 0.09 -3.98 0.000262

DMN-b PFCd 5 0.540 ± 0.087 0.454 ± 0.089 0.10 -3.79 0.000458

DMN-a PFCm 1 0.536 ± 0.065 0.471 ± 0.066 0.07 -3.65 0.000713

DMN-a PFCd 3 0.525 ± 0.073 0.469 ± 0.071 0.07 -3.53 0.001002

DMN-b PFCd 1 0.543 ± 0.065 0.471 ± 0.075 0.08 -3.49 0.001117

SomMoTA 16 0.546 ± 0.089 0.458 ± 0.104 0.10 -3.42 0.001383

Hippocampus 0.501 ± 0.094 0.429 ± 0.081 0.09 -3.35 0.001687

SomMoTA 12 0.544 ± 0.089 0.466 ± 0.089 0.09 -3.29 0.002004

RH 

DMN-a PFCm 1 0.532 ± 0.062 0.451 ± 0.071 0.09 -4.77 0.000022

DMN-a PFCd 2 0.559 ± 0.061 0.466 ± 0.080 0.10 -4.46 0.000058

Hippocampus 0.514 ± 0.123 0.421 ± 0.093 0.11 -3.78 0.000472

DMN-a PFCm 4 0.542 ± 0.062 0.477 ± 0.071 0.07 -3.52 0.001023

DMN-b Ant Temp 1 0.541 ± 0.063 0.467 ± 0.096 0.09 -3.51 0.001052

DMN-b PFCd 4 0.558 ± 0.066 0.488 ± 0.101 0.09 -3.46 0.001215

DMN-a PCC 1 0.573 ± 0.057 0.512 ± 0.069 0.07 -3.46 0.001222

DMN-a PFCm 3 0.507 ± 0.086 0.452 ± 0.075 0.08 -3.43 0.001335

SalVentAttn A ParMed7 0.497 ± 0.067 0.437 ± 0.057 0.06 -3.34 0.001735

* p-values are uncorrected 2-sided

Any inconsistencies in the S1 file have also been rectified.

As recommended, we have replaced the column header ‘statistic’ with ‘t-statistic’ in the relevant tables throughout the main manuscript and the S1 file.

Reviewer #2: 

The authors addressed all the comments adequately, and their manuscript is significantly improved. I have no further comments.

---

## [Editor Report · Decision Letter 2]

12 Sep 2019

No relationship between fornix and cingulum degradation and within-network decreases in functional connectivity in prodromal Alzheimer’s disease

PONE-D-19-16445R2

Dear Dr. Gilligan,

We are pleased to inform you that your manuscript has been judged scientifically suitable for publication and will be formally accepted for publication once it complies with all outstanding technical requirements.

With kind regards,

Han Zhang, Ph.D

Academic Editor

PLOS ONE

---

## [Editor Report · Acceptance letter]

25 Sep 2019

PONE-D-19-16445R2 

No relationship between fornix and cingulum degradation and within-network decreases in functional connectivity in prodromal Alzheimer’s disease 

Dear Dr. Gilligan:

I am pleased to inform you that your manuscript has been deemed suitable for publication in PLOS ONE. Congratulations! Your manuscript is now with our production department. 

With kind regards,

on behalf of

Dr. Han Zhang 

Academic Editor

PLOS ONE